# On the Tightness of Semidefinite Relaxations for Certifying Robustness to Adversarial Examples

**Richard Y. Zhang**
Department of Electrical and Computer Engineering,
University of Illinois at Urbana-Champaign,
Urbana, 61801 IL, USA.
`ryz@illinois.edu`

## Abstract

The robustness of a neural network to adversarial examples can be provably certified by solving a convex relaxation. If the relaxation is loose, however, then the resulting certificate can be too conservative to be practically useful. Recently, a less conservative robustness certificate was proposed, based on a semidefinite programming (SDP) relaxation of the ReLU activation function. In this paper, we describe a geometric technique that determines whether this SDP certificate is *exact*, meaning whether it provides both a lower-bound on the size of the smallest adversarial perturbation, as well as a globally optimal perturbation that attains the lower-bound. Concretely, we show, for a least-squares restriction of the usual adversarial attack problem, that the SDP relaxation amounts to the nonconvex projection of a point onto a hyperbola. The resulting SDP certificate is exact if and only if the projection of the point lies on the major axis of the hyperbola. Using this geometric technique, we prove that the certificate is exact over a single hidden layer under mild assumptions, and explain why it is usually conservative for several hidden layers. We experimentally confirm our theoretical insights using a general-purpose interior-point method and a custom rank-2 Burer-Monteiro algorithm.

## 1   Introduction

It is now well-known that neural networks are vulnerable to *adversarial examples*: imperceptibly small changes to the input that result in large, possibly targeted change to the output [1–3]. Adversarial examples are particularly concerning for safety-critical applications like self-driving cars and smart grids, because they present a mechanism for erratic behavior and a vector for malicious attacks.

Methods for analyzing robustness to adversarial examples work by formulating the problem of finding the *smallest* perturbation needed to result in an adversarial outcome. For example, this could be the smallest change to an image of the digit "3" for a given model to misclassify it as an "8". The *size* of this smallest change serves as a *robustness margin*: the model is robust if even the smallest adversarial change is still easily detectable.

Computing the robustness margin is a nonconvex optimization problem. In fact, methods that *attack* a model work by locally solving this optimization, usually using a variant of gradient descent [3–6]. A successful attack demonstrates vulnerability by explicitly stating a small—but not necessarily the smallest—adversarial perturbation. Of course, failed attacks do not prove robustness, as there is always the risk of being defeated by stronger attacks in the future. Instead, robustness can be *certified* by proving lower-bounds on the robustness margin [7–18]. Training against a robustness certificate (as an adversary) in turn produces models that are certifiably robust to adversarial examples [10, 19, 20].

The most useful robustness certificates are *exact*, meaning that they also explicitly state an adversarial perturbation whose size matches their lower-bound on the robustness margin, thereby proving global optimality [7–9]. Unfortunately, the robustness certification problem is NP-hard in general, so most existing methods for exact certification require worst-case time that scales exponentially with respect to the number of neurons. In contrast, *conservative* certificates are more scalable because the have polynomial worst-case time complexity [10–18]. Their usefulness is derived from their level of conservatism. The issue is that a pessimistic assessement for a model that is ostensibly robust can be attributed to either undue conservatism in the certificate, or an undiscovered vulnerability in the model. Also, training against an overly conservative certificate will result in an overly cautious model that is too willing to sacrifice performance for perceived safety.

Recently, Raghunathan et al. [21] proposed a less conservative certificate based on a *semidefinite programming* (SDP) relaxation of the rectified linear unit (ReLU) activation function. Their empirical results found it to be significantly less conservative than competing approaches, based on linear programming or propagating Lipschitz constants. In other domains, ranging from integer programming [22, 23], polynomial optimization [24, 25], matrix completion [26, 27], to matrix sensing [28], the SDP relaxation is often *tight*, in the sense that it is formally equivalent to the original combinatorially hard problem. Within our context, tightness corresponds to exactness in the robustness certificate. Hence, the SDP relaxation is a good candidate for exact certification in polynomial time, possibly over some restricted class of models or datasets.

This paper aims to understand when the SDP relaxation of the ReLU becomes tight, with the goal of characterizing conditions for exact robustness certification. Our main contribution is a geometric technique for analyzing tightness, based on splitting a least-squares restriction of the adversarial attack problem into a sequence of projection problems. The final problem projects a point onto a nonconvex hyperboloid (i.e. a high-dimensional hyperbola), and the SDP relaxation is tight if and only if this projection lies on the major axis of the hyperboloid. Using this geometric technique, we prove that the SDP certificate is generally exact for a single hidden layer. The certificate is usually conservative for several hidden layers; we use the same geometric technique to offer an explanation for why this is the case.

**Notations.** Denote *vectors* in boldface lower-case $\mathbf{x}$, *matrices* in boldface upper-case $\mathbf{X}$, and *scalars* in non-boldface $x, X$. The bracket denotes *indexing* $\mathbf{x}[i]$ starting from 1, and also *concatenation*, which is row-wise via the comma $[a, b]$ and column-wise via the semicolon $[a; b]$. The $i$-th canonical basis vector $\mathbf{e}_i$ satisfies $\mathbf{e}_i[i] = 1$ and $\mathbf{e}_i[j] = 0$ for all $j \neq i$. The usual *inner product* is $\langle \mathbf{a}, \mathbf{b} \rangle = \sum_i \mathbf{a}[i]\mathbf{b}[i]$, and the usual rectified linear unit activation function is $\mathrm{ReLU}(\mathbf{x}) \equiv \max\{\mathbf{x}, 0\}$.

## 2 Main results

Let $\mathbf{f} : \mathbb{R}^n \to \mathbb{R}^m$ be a feedforward ReLU neural network classifier with $\ell$ hidden layers

$$\mathbf{f}(\mathbf{x}_0) = \mathbf{x}_\ell \text{ where } \mathbf{x}_{k+1} = \mathrm{ReLU}(\mathbf{W}_k\mathbf{x}_k + \mathbf{b}_k) \quad \text{for all } k \in \{0, 1, \ldots, \ell-1\}, \quad (2.1)$$

that takes an input $\hat{\mathbf{x}} \in \mathbb{R}^n$ (say, an image of a hand-written single digit) labeled as belonging to the $i$-th of $m$ classes (say, the 5-th of 10 possible classes of single digits), and outputs a prediction vector $\mathbf{s} = \mathbf{W}_\ell\mathbf{f}(\hat{\mathbf{x}}) + \mathbf{b}_\ell \in \mathbb{R}^m$ whose $i$-th element is the largest, as in $\mathbf{s}[i] > \mathbf{s}[j]$ for all $j \neq i$. Then, the problem of finding an adversarial example $\mathbf{x}$ that is similar to $\hat{\mathbf{x}}$ but causes an incorrect $j$-th class to be ranked over the $i$-th class can be posed

$$d_j^\star = \min_{\mathbf{x} \in \mathbb{R}^n} \|\mathbf{x} - \hat{\mathbf{x}}\| \quad \text{subject to} \quad (2.1), \quad \langle \mathbf{w}, \mathbf{f}(\mathbf{x}) \rangle + b \leq 0, \quad (\mathrm{A})$$

where $\mathbf{w} = \mathbf{W}_\ell^T(\mathbf{e}_i - \mathbf{e}_j)$ and $b = \mathbf{b}_\ell^T(\mathbf{e}_i - \mathbf{e}_j)$. In turn, the adversarial example $\mathbf{x}^\star$ most similar to $\hat{\mathbf{x}}$ over *all* incorrect classes gives a *robustness margin* $d^\star = \min_{j \neq i} d_j$ for the neural network.

In practice, the SDP relaxation for problem (A) is often loose. To understand the underlying mechanism, we study a slight modification that we call its *least-squares restriction*

$$L^\star = \min_{\mathbf{x} \in \mathbb{R}^n} \|\mathbf{x} - \hat{\mathbf{x}}\| \quad \text{subject to} \quad (2.1), \quad \|\mathbf{f}(\mathbf{x}) - \hat{\mathbf{z}}\| \leq \rho, \quad (\mathrm{B})$$

where $\hat{\mathbf{z}} \in \mathbb{R}^m$ is the targeted output, and $\rho > 0$ is a radius parameter. Problem (A) is equivalent to problem (B) taken at the limit $\rho \to \infty$, because a half-space is just an infinite-sized ball

$$\|\mathbf{z} + \mathbf{w}\left(b/\|\mathbf{w}\|^2 + \rho/\|\mathbf{w}\|\right)\|^2 \leq \rho^2 \quad \Longleftrightarrow \quad \frac{\|\mathbf{w}\|}{2\rho}\|\mathbf{z} + \mathbf{w}\left(b/\|\mathbf{w}\|^2\right)\|^2 + [\langle \mathbf{w}, \mathbf{z} \rangle + b] \leq 0 \quad (2.2)$$

with a center $\hat{\mathbf{z}} = -\mathbf{w} \left( b/\|\mathbf{w}\|^2 + \rho/\|\mathbf{w}\| \right)$ that tends to infinity alongside the ball radius $\rho$. The SDP relaxation for problem (B) is often tight for finite values of the radius $\rho$. The resulting solution $\mathbf{x}$ is a *strictly* feasible (but suboptimal) attack for problem (A) that causes misclassification $\langle \mathbf{w}, \mathbf{f}(\mathbf{x}) \rangle + b < 0$. The corresponding optimal value $L^\star$ gives an upper-bound $d_{\text{ub}} \equiv L^\star \geq d^\star$ that converges to an equality as $\rho \to \infty$. (See Appendix E for details.)

In Section 5, we completely characterize the SDP relaxation for problem (B) over a single hidden neuron, by appealing to the underlying geometry of the relaxation. In Section 6, we extend these insights to partially characterize the case of a single hidden layer.

**Theorem 2.1** (One hidden neuron)**.** *Consider the one-neuron version of problem (B), explicitly written*

$$L^\star \quad = \quad \min_x \quad |x - \hat{x}| \quad subject\ to \quad |\text{ReLU}(x) - \hat{z}| \leq \rho. \tag{2.3}$$

*The SDP relaxation of (2.3) yields a tight lower-bound $L_{\text{lb}} = L^\star$ and a globally optimal $x^\star$ satisfying $|x^\star - \hat{x}| = L_{\text{lb}}$ if and only if one of the two conditions hold: (i) $\rho \geq |\hat{z}|$; or (ii) $\rho < \hat{z}/(1 - \min\{0, \hat{x}/\hat{z}\})$.*

**Theorem 2.2** (One hidden layer)**.** *Consider the one-layer version of problem (B), explicitly written*

$$L^\star \quad = \quad \min_{\mathbf{x} \in \mathbb{R}^n} \quad \|\mathbf{x} - \hat{\mathbf{x}}\| \quad s.t. \quad \|\text{ReLU}(\mathbf{W}\mathbf{x}) - \hat{\mathbf{z}}\| \leq \rho \tag{2.4}$$

*The SDP relaxation of (2.3) yields a tight lower-bound $L_{\text{lb}} = L^\star$ and a globally optimal $\mathbf{x}^\star$ satisfying $\|\mathbf{x}^\star - \hat{\mathbf{x}}\| = L_{\text{lb}}$ if one of the two conditions hold: (i) $\rho \geq \|\text{ReLU}(\mathbf{W}\hat{\mathbf{x}}) - \hat{\mathbf{z}}\|$; or (ii) $\rho < \hat{z}_{\min}/2(1 + \kappa)$ and $\|\mathbf{W}\hat{\mathbf{x}} - \hat{\mathbf{z}}\|_\infty < \hat{z}_{\min}^2/(2\rho\kappa)$ where $\hat{z}_{\min} = \min_i \hat{z}_i$ and $\kappa = \|\mathbf{W}\|^2 \|(\mathbf{W}\mathbf{W}^T)^{-1}\|_\infty$.*

The lack of a weight term in (2.3) and a bias term in (2.3) and (2.4) is without loss of generality, as these can always be accommodated by shifting and scaling $\mathbf{x}$ and $\hat{\mathbf{x}}$. Intuitively, Theorem 2.1 and Theorem 2.2 say that the SDP relaxation tends to be tight if the output target $\hat{\mathbf{z}}$ is *feasible*, meaning that there exists some choice of $\mathbf{u}$ such that $\hat{\mathbf{z}} = \mathbf{f}(\mathbf{u})$. (The condition $\rho < \hat{z}_{\min}/2(1 + \kappa)$ is sufficient for feasibility.) Conversely, the SDP relaxation tends to be loose if the radius $\rho > 0$ lies within an intermediate band of "bad" values. For example, over a single neuron with a feasible $\hat{z} = 1$, the relaxation is loose if and only if $\hat{x} \leq 0$ and $1/(1 + |\hat{x}|) \leq \rho < 1$. These two general trends are experimentally verified in Section 8.

In the case of multiple layers, the SDP relaxation is usually loose, with a notable exception being the trivial case with $L^\star = 0$.

**Corollary 2.3** (Multiple layers)**.** *If $\rho \geq \|\mathbf{f}(\hat{\mathbf{x}}) - \hat{\mathbf{z}}\|$, then the SDP relaxation of problem (B) yields the tight lower-bound $L_{\text{lb}} = L^\star = 0$ and the globally optimal $\mathbf{x}^\star = \hat{\mathbf{x}}$ satisfying $\|\mathbf{x}^\star - \hat{\mathbf{x}}\| = 0$.*

The proof is given in Appendix E. In Section 7, we explain the looseness of the relaxation for multiple layers using the geometric insight developed for the single layer. As mentioned above, the general looseness of the SDP relaxation for problem (B) then immediately implies the general looseness for problem (A).

## 3 Related work

**Adversarial attacks, robustness certificates, and certifiably robust models.** Adversarial examples are usually found by using projected gradient descent to solve problem (A) with its objective and constraint swapped [3–6]. Training a model against these empirical attacks generally yield very resilient models [4–6]. It is possible to certify robustness exactly despite the NP-hardness of the problem [7–9, 29]. Nevertheless, conservative certificates show greater promise for scalability because they are polynomial-time algorithms. From the perspective of tightness, the next most promising techniques after the SDP relaxation are relaxations based on linear programming (LP) [10–13], though techniques based on propagating bounds and/or Lipschitz constants tend to be much faster in practice [14–18]. Aside from training a model against a robustness certificate [10, 19, 20], certifiably robust models can also be constructed by randomized smoothing [30–32].

**Tightness of SDP relaxations.** The geometric techniques used in our analysis are grounded in the classic paper of Goemans and Williamson [33] (see also [34–36]), but our focuses are different: they prove general bounds valid over entire classes of SDP relaxations, whereas we identify specific SDP relaxations that are exactly tight. In the sense of tight relaxations, our results are reminiscent of

the guarantees by Candès and Recht [26], Candès and Tao [27] (see also [37, 38]) on the matrix completion problem, but our approaches are very different: their arguments are based on using the dual to imply tightness in the primal, whereas our proof analyzes the primal directly.

After this paper was submitted, we became aware of two parallel work [39, 40] that also study the tightness of SDP relaxations for robustness to adversarial examples. The first, due to Fazlyab et al. [39], uses similar techniques like the S-procedure to study a different SDP relaxation constructed from robust control techniques. The second, due to Dvijotham et al. [40], studies the same SDP relaxation within the context of a different attack problem, namely the version of Problem (A) with the objective replaced by the infinity norm distance $\|\mathbf{x} - \hat{\mathbf{x}}\|_\infty$.

**Efficient algorithms for SDPs.** While SDPs are computationally expensive to solve using off-the-shelf algorithms, efficient formulation-specific solvers were eventually developed once their use case became sufficiently justified. In fact, most state-of-the-art algorithms for phase retrieval [41–44] and collaborative filtering [45–49] can be viewed as highly optimized algorithms to solve an underlying SDP relaxation. Our rank-2 Burer-Monteiro algorithm in Section 8 is inspired by Burer et al. [50]. It takes strides towards an efficient algorithm, but the primary focus of this paper is to understand the use case for robustness certification.

## 4 Preliminary: Geometry of the SDP relaxation

The SDP relaxation of Raghunathan et al. [21] is based on the observation that the rectified linear unit (ReLU) activation function $z = \text{ReLU}(x) \equiv \max\{0, x\}$ is equivalent to the inequalities $z \geq 0$, $z \geq x$, and $z(z - x) \leq 0$. Viewing these as quadratics, we apply a standard technique (see Shor [51] and also [24, 25, 33]) to rewrite them as linear inequalities over a positive semidefinite matrix variable,

$$z \geq 0, \quad z \geq x, \quad Z \leq Y, \quad \mathbf{G} = \begin{bmatrix} 1 & x & z \\ x & X & Y \\ z & Y & Z \end{bmatrix} \succeq 0, \quad \text{rank}(\mathbf{G}) = 1. \tag{4.1}$$

In essence, the reformulation collects the inherent nonconvexity of $\text{ReLU}(\cdot)$ into the constraint $\text{rank}(\mathbf{G}) = 1$, which can then be deleted to yield a convex relaxation. If the relaxation has a unique solution $\mathbf{G}^\star$ satisfying $\text{rank}(\mathbf{G}^\star) = 1$, then we say that it is *tight*.[1] In this case, the globally optimal solution $x^\star, z^\star$ to the original nonconvex problem can be found by solving the SDP relaxation in polynomial time and factorizing the solution $\mathbf{G}^\star = \mathbf{g}\mathbf{g}^T$ where $\mathbf{g} = [1; x^\star; z^\star]^T$.

It is helpful to view $\mathbf{G}$ as the *Gram matrix* associated with the vectors $\mathbf{e}, \mathbf{x}, \mathbf{z} \in \mathbb{R}^p$ in an ambient $p$-dimensional space, where $p$ is the order of $\mathbf{G}$ (here $p = 3$). The individual elements of $\mathbf{G}$ correspond to the inner products terms associated with $\mathbf{e}, \mathbf{x}, \mathbf{z}$, as in

$$\langle \mathbf{e}, \mathbf{z} \rangle \geq \max\{0, \langle \mathbf{e}, \mathbf{x} \rangle\}, \quad \|\mathbf{z}\|^2 \leq \langle \mathbf{z}, \mathbf{x} \rangle, \quad \|\mathbf{e}\|^2 = 1, \quad \mathbf{G} = \begin{bmatrix} \langle \mathbf{e}, \mathbf{e} \rangle & \langle \mathbf{e}, \mathbf{x} \rangle & \langle \mathbf{e}, \mathbf{z} \rangle \\ \langle \mathbf{e}, \mathbf{x} \rangle & \langle \mathbf{x}, \mathbf{x} \rangle & \langle \mathbf{x}, \mathbf{z} \rangle \\ \langle \mathbf{e}, \mathbf{z} \rangle & \langle \mathbf{x}, \mathbf{z} \rangle & \langle \mathbf{z}, \mathbf{z} \rangle \end{bmatrix}, \tag{4.2}$$

and $\text{rank}(\mathbf{G}) = 1$ corresponds to *collinearity* between $\mathbf{x}, \mathbf{z},$ and $\mathbf{e}$, as in $\|\mathbf{e}\|\|\mathbf{x}\| = |\langle \mathbf{e}, \mathbf{x} \rangle|$ and $\|\mathbf{e}\|\|\mathbf{z}\| = |\langle \mathbf{e}, \mathbf{z} \rangle|$. From the Gram matrix perspective, the SDP relaxation works by allowing the underlying vectors $\mathbf{x}, \mathbf{z},$ and $\mathbf{e}$ to take on arbitrary directions; the relaxation is tight if and only if *all* possible solutions $\mathbf{e}^\star, \mathbf{x}^\star, \mathbf{z}^\star$ are collinear.

Figure 1 shows the underlying geometry the ReLU constraints (4.2) as noted by Raghunathan et al. [21]. Take $\mathbf{z}$ as the variable and fix $\mathbf{e}, \mathbf{x}$. Since $\mathbf{e}$ is a unit vector, we may view $\langle \mathbf{e}, \mathbf{x} \rangle$ and $\langle \mathbf{e}, \mathbf{z} \rangle$ as the "e-axis coordinates" for the vectors $\mathbf{x}$ and $\mathbf{z}$. The constraint $\langle \mathbf{e}, \mathbf{z} \rangle \geq \max\{0, \langle \mathbf{e}, \mathbf{x} \rangle\}$ is then a *half-space* that restricts the "e-coordinate" of $\mathbf{z}$ to be nonnegative and greater than that of $\mathbf{x}$. The constraint $\langle \mathbf{z}, \mathbf{z} - \mathbf{x} \rangle \leq 0$ is rewritten as $\|\mathbf{z} - \mathbf{x}/2\|^2 \leq \|\mathbf{x}/2\|^2$ by completing the square; this is a *sphere* that restricts $\mathbf{z}$ to lie within a distance of $\|\mathbf{x}/2\|$ from the center $\mathbf{x}/2$. Combined, the ReLU constraints (4.2) constrain $\mathbf{z}$ to lie within a *spherical cap*—a portion of a sphere cut off by a plane.

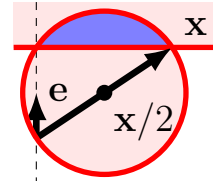

**Fig. 1** – The ReLU constraints (4.2) describe a spherical cap.

# 5 Tightness for one hidden neuron

Now, consider the SDP relaxation of the one-neuron problem (2.3), explicitly written as

$$L_{\mathrm{lb}}^2 = \min_{\mathbf{G}} \quad X - 2x\hat{x} + \hat{x}^2 \quad \text{s.t.} \quad \begin{aligned} z &\geq \max\{0, x\}, \ Z \leq Y, \\ Z &- 2z\hat{z} + \hat{z}^2 \leq \rho^2, \end{aligned} \quad \mathbf{G} = \begin{bmatrix} 1 & x & z \\ x & X & Y \\ z & Y & Z \end{bmatrix} \succeq 0. \quad (5.1)$$

Viewing the matrix variable $\mathbf{G} \succeq 0$ as the Gram matrix associated with the vectors $\mathbf{e}, \mathbf{x}, \mathbf{z} \in \mathbb{R}^p$ where $p = 3$ rewrites (5.1) as the following

$$L_{\mathrm{lb}} = \min_{\mathbf{x}, \mathbf{z}, \mathbf{e} \in \mathbb{R}^p} \|\mathbf{x} - \hat{x}\, \mathbf{e}\| \quad \text{s.t.} \ \langle \mathbf{z}, \mathbf{e} \rangle \geq \max\{\langle \mathbf{x}, \mathbf{e} \rangle, 0\}, \ \|\mathbf{z}\|^2 \leq \langle \mathbf{z}, \mathbf{x} \rangle, \ \|\mathbf{z} - \hat{z}\, \mathbf{e}\| \leq \rho. \quad (5.2)$$

The SDP relaxation (5.1) has a unique rank-1 solution if and only if its nonconvex vector interpretation (5.2) has a unique solution that aligns with $\mathbf{e}$. The proof for the following is given in Appendix A.

**Lemma 5.1** (Collinearity and rank-1). *Fix $\mathbf{e} \in \mathbb{R}^p$. Then, problem (5.2) has a unique solution $\mathbf{x}^\star$ satisfying $\|\mathbf{x}^\star\| = |\langle \mathbf{x}^\star, \mathbf{e} \rangle|$ if and only if problem (5.1) has a unique solution $\mathbf{G}^\star$ satisfying $\mathrm{rank}(\mathbf{G}^\star) = 1$.*

We proceed to solve problem (5.2) by rewriting it as the composition of a convex projection over $\mathbf{z}$ with a nonconvex projection over $\mathbf{x}$, as in:

$$\phi(\mathbf{x}, \hat{z}) = \min_{\mathbf{z} \in \mathbb{R}^p} \|\mathbf{z} - \hat{z}\mathbf{e}\| \quad \text{subject to} \quad \langle \mathbf{e}, \mathbf{z} \rangle \geq \max\{\langle \mathbf{e}, \mathbf{x} \rangle, 0\}, \quad \|\mathbf{z}\|^2 \leq \langle \mathbf{z}, \mathbf{x} \rangle, \quad (5.3)$$

$$L_{\mathrm{lb}} = \min_{\mathbf{x} \in \mathbb{R}^p} \|\mathbf{x} - \hat{x}\mathbf{e}\| \quad \text{subject to} \quad \phi(\mathbf{x}, \hat{z}) \leq \rho. \quad (5.4)$$

Problem (5.3) is clearly the projection of the point $\hat{z}\mathbf{e}$ onto the spherical cap shown in Figure 1. In a remarkable symmetry, it turns out that problem (5.4) is the projection of the point $\hat{x}\mathbf{e}$ onto a *hyperboloidal cap*—a portion of a high-dimensional hyperbola cut off by a plane—with the optimal $\mathbf{x}^\star$ in problem (5.2) being the resulting projection. In turn, our goal of verifying collinearity between $\mathbf{x}^\star$ and $\mathbf{e}$ amounts to checking whether $\hat{x}\mathbf{e}$ projects onto the major axis of the hyperboloid.

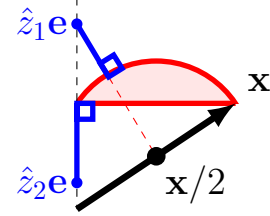

**Fig. 2** – Problem (5.3) is the projection of a point onto a spherical cap.

To turn this intuitive sketch into a rigorous proof, we begin by solving the convex projection (5.3) onto the spherical cap. Figure 2 shows the corresponding geometry. There are two distinct scenarios: **i)** For $\hat{z}_1\mathbf{e}$ that is *above* the spherical cap, the projection must intersect the upper, round portion of the spherical cap along the line from $\hat{z}_1\mathbf{e}$ to $\mathbf{x}/2$. This yields a distance of $\phi(\mathbf{x}, \hat{z}_1) = \|\hat{z}_1\mathbf{e} - \mathbf{x}/2\| - \|\mathbf{x}/2\|$. **ii)** For $\hat{z}_2\mathbf{e}$ that is *below* the spherical cap, the projection is simply the closest point directly above, at a distance of $\phi(\mathbf{x}, \hat{z}_2) = \max\{0, \langle \mathbf{e}, \mathbf{x} \rangle\} - \hat{z}_2$.

It turns out that the conditional statements are unnecessary; the distance $\phi(\mathbf{x}, \hat{z})$ simply takes on the larger of the two values derived above. In Appendix B, we prove this claim algebraically, thereby establishing the following.

**Lemma 5.2** (Projection onto spherical cap). *The function $\phi : \mathbb{R}^p \times \mathbb{R} \to \mathbb{R}$ defined in (5.3) satisfies*

$$\phi(\mathbf{x}, \hat{z}) = \max\{\max\{0, \langle \mathbf{e}, \mathbf{x} \rangle\} - \hat{z}, \quad \|\hat{z}\mathbf{e} - \mathbf{x}/2\| - \|\mathbf{x}/2\|\}. \quad (5.5)$$

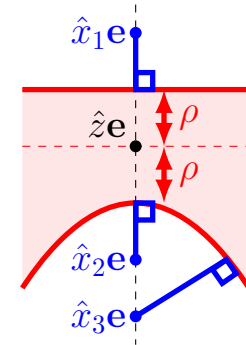

**Fig. 3** – Problem (5.4) is the projection of a point onto a hyperbolidal cap.

Taking $\mathbf{x}$ as the variable, we see from Lemma 5.2 that each level set $\phi(\mathbf{x}, \hat{z}) = \rho$ is either: 1) a *hyperplane* normal to $\mathbf{e}$ at the intercept $\hat{z} + \rho$; or 2) a two-sheet *hyperboloid* centered at $\hat{z}\mathbf{e}$, with semi-major axis $\rho$ and focal distance $|\hat{z}|$ in the direction of $\mathbf{e}$. Hence, the sublevel set $\phi(\mathbf{x}, \hat{z}) \leq \rho$ is a hyperboloidal cap as claimed.

We proceed to solve the nonconvex projection (5.4) onto the hyperboloidal cap. This shape degenerates into a half-space if the semi-major axis $\rho$ is longer than the focal distance, as in $\rho \geq |\hat{z}|$, and becomes empty altogether with a negative center, as in $\hat{z} < -\rho$. Figure 3 shows the geometry of projecting onto a *nondegenerate* hyperboloidal cap with $\hat{z} > \rho$. There are three distinct scenarios: **i)** For $\hat{x}_1\mathbf{e}$ that is

either *above* or *interior* to the hyperboloidal cap, the projection is either the closest point directly below or the point itself, as in $\mathbf{x}^\star = \min\{\hat{z} + \rho, \hat{x}_1\}\mathbf{e}$; **ii)** For $\hat{x}_2\mathbf{e}$ that is *below* and *sufficiently close* to the hyperboloidal cap, the projection lies at the top of the hyperbolid sheet at $\mathbf{x}^\star = (\hat{z} - \rho)\mathbf{e}$; **iii)** For $\hat{x}_3\mathbf{e}$ that is *below* and *far away* from the hyperboloidal cap, the projection lies somewhere along the side of the hyperboloid.

Evidently, the first two scenarios correspond to choices of $\mathbf{x}^\star$ that are collinear to $\mathbf{e}$, while the third scenario does not. To resolve the boundary between the second and third scenarios, we solve the projection onto a hyperbolidal cap in closed-form.

**Lemma 5.3** (Projection onto nondegenerate hyperboloidal cap)**.** *Given* $\mathbf{e} \in \mathbb{R}^p$, $\hat{x} \in \mathbb{R}$, *and* $\hat{z} > \rho > 0$, *define* $\mathbf{x}^\star$ *as the solution to the following projection*

$$\min_{\mathbf{x} \in \mathbb{R}^p} \quad \|\mathbf{x} - \hat{x}\mathbf{e}\|^2 \quad s.t. \quad \langle \mathbf{e}, \mathbf{x} \rangle - \hat{z} \leq \rho, \quad \|\hat{z}\mathbf{e} - \mathbf{x}/2\| - \|\mathbf{x}/2\| \leq \rho.$$

*Then,* $\mathbf{x}^\star$ *is unique and satisfies* $\|\mathbf{x}^\star\| = |\langle \mathbf{e}, \mathbf{x}^\star \rangle|$ *if and only if* $(\hat{z} - \hat{x}) < \hat{z}^2/\rho$.

We defer the proof of Lemma 5.3 to Appendix C, but note that the main idea is to use the S-lemma (see e.g. [52, p. 655] or [53]) to solve the minimization of a quadratic (the distance) subject to a quadratic constraint (the nondegenerate hyperboloid). Resolving the degenerate cases and applying Lemma 5.3 to the nondegenerate case yields a proof of our main result.

*Proof of Theorem 2.1.* If $\hat{z} < -\rho$, then the hyperbolidal cap $\phi(\mathbf{x}, \hat{z}) \leq \rho$ is empty as $\rho < -\hat{z} \leq \phi(\mathbf{x}, \hat{z})$. In this case, problem (5.4) is infeasible. If $|\hat{z}| \leq \rho$, then the hyperbolidal cap $\phi(\mathbf{x}, \hat{z}) \leq \rho$ degenerates into a half-space $\langle \mathbf{e}, \mathbf{x} \rangle \leq \hat{z} + \rho$, because $\|\hat{z}\mathbf{e} - \mathbf{x}/2\| - \|\mathbf{x}/2\| \leq \|\hat{z}\mathbf{e}\| + \|\mathbf{x}/2\| - \|\mathbf{x}/2\| = |\hat{z}| \leq \rho$. In this case, the projection $\mathbf{x}^\star = \min\{\hat{z} + \rho, \hat{x}\}\mathbf{e}$ is clearly collinear to $\mathbf{e}$, so $|\hat{z}| \leq \rho$ is the first condition of Theorem 2.1. Finally, if $\hat{z} > \rho$, Lemma 5.3 says that $\mathbf{x}^\star$ is collinear with $\mathbf{e}$ whenever $(\hat{z} - \hat{x}) < \hat{z}^2/\rho$, which is rewritten $\hat{x} > \hat{z}(1 - \hat{z}/\rho)$. Under $\hat{z} > \rho$, this is equivalent to $\rho < \hat{z}/(1 - \hat{x}/\hat{z})$. Finally, taking the intersection of these two constraints yields $\rho < \hat{z}/(1 - \min\{0, \hat{x}/\hat{z}\})$, which is the second condition of Theorem 2.1. $\qquad\square$

# 6 Tightness for one layer

Our analysis of the one-hidden-neuron case extends to the one-hidden-layer case without significant modification. Here, the semidefinite relaxation reads

$$L_{\text{lb}}^2 = \min_{\mathbf{G}} \quad \text{tr}(\mathbf{X}) - 2\langle \mathbf{x}, \hat{\mathbf{x}} \rangle + \|\hat{\mathbf{x}}\|^2 \tag{6.1}$$

$$\text{s.t.} \quad \begin{aligned} &\mathbf{z} \geq \max\{0, \mathbf{Wx}\}, \ \text{diag}(\mathbf{WZ}) \leq \text{diag}(\mathbf{WY}), \\ &\text{tr}(\mathbf{Z}) - 2\langle \mathbf{z}, \hat{\mathbf{z}} \rangle + \|\hat{\mathbf{z}}\|^2 \leq \rho^2, \end{aligned} \quad \mathbf{G} = \begin{bmatrix} 1 & \mathbf{x} & \mathbf{z} \\ \mathbf{x} & \mathbf{X} & \mathbf{Y} \\ \mathbf{z} & \mathbf{Y}^T & \mathbf{Z} \end{bmatrix} \succeq 0.$$

Viewing the matrix variable $\mathbf{G} \succeq 0$ in the corresponding SDP relaxation (6.1) as the Gram matrix associated a set of length-$p$ vectors (where $p = m + n + 1$ is the order of the matrix $\mathbf{G}$) yields the following[2]

$$L_{\text{lb}}^2 = \min_{\mathbf{x}_j, \mathbf{z}_i \in \mathbb{R}^p} \quad \sum_j \|\mathbf{x}_j - \hat{x}_j\, \mathbf{e}\|^2 \tag{6.2}$$

$$\text{s.t.} \quad \begin{aligned} &\langle \mathbf{e}, \mathbf{z}_i \rangle \geq \max\left\{0, \langle \mathbf{e}, \textstyle\sum_j W_{i,j}\mathbf{x}_j \rangle\right\}, \\ &\|\mathbf{z}_i\|^2 \leq \langle \mathbf{z}_i, \textstyle\sum_j W_{i,j}\mathbf{x}_j \rangle, \end{aligned} \quad \textstyle\sum_i \|\mathbf{z}_i - \hat{z}_i\, \mathbf{e}\|^2 \leq \rho^2 \text{ for all } i,$$

with indices $i \in \{1, 2, \ldots, m\}$ and $j \in \{1, 2, \ldots, n\}$. We will derive conditions for the SDP relaxation (6.1) to have a unique, rank-1 solution by fixing $\mathbf{e}$ in problem (6.2) and verifying that every optimal $\mathbf{x}_j^\star$ is collinear with $\mathbf{e}$ for all $j$. The proof for the following is given in Appendix A.

**Lemma 6.1** (Collinearity and rank-1)**.** *Fix* $\mathbf{e} \in \mathbb{R}^p$*. Then, problem (6.2) has a unique solution* $\mathbf{x}_1^\star, \mathbf{x}_2^\star, \ldots, \mathbf{x}_n^\star$ *satisfying* $\|\mathbf{x}_j^\star\| = |\langle \mathbf{x}_j^\star, \mathbf{e} \rangle|$ *if and only if problem (6.1) has a unique solution* $\mathbf{G}^\star$ *satisfying* $\text{rank}(\mathbf{G}^\star) = 1$.

Problem (6.2) can be rewritten as the composition of a series of projections over $\mathbf{z}_i$, followed by a sequence of nonconvex projections over $\mathbf{x}_j$, as in

$$L_{\text{lb}}^2 = \min_{\mathbf{x}_j \in \mathbb{R}^p, a_i \geq 0} \quad \sum_j \|\mathbf{x}_j - \hat{x}_j \mathbf{e}\|^2 \quad \text{s.t.} \quad \phi(\textstyle\sum_j W_{i,j} \mathbf{x}_j, \hat{z}_i) \leq \rho_i \text{ for all } i, \quad \sum_i \rho_i^2 \leq \rho^2, \quad (6.3)$$

where $\phi$ was previously defined in the one-neuron convex projection (5.3). Whereas in the one-neuron case we are projecting a single point onto a single hyperboloidal cap, the one-layer case requires us to project $n$ points onto the intersection of $n$ hyperboloidal caps. This has a closed-form solution only when all the hyperboloids are nondegenerate.

**Lemma 6.2** (Projection onto several hyperboloidal caps). *Given* $\mathbf{W} = [W_{i,j}] \in \mathbb{R}^{m \times n}$, $\hat{\mathbf{x}} = [\hat{x}_j] \in \mathbb{R}^n$, $\hat{\mathbf{z}} = [\hat{z}_i] \in \mathbb{R}^m$, $\mathbf{e} \in \mathbb{R}^p$, *and* $\rho_i$ *satisfying* $\hat{z}_i > \rho > 0$, *define* $\mathbf{x}_j^\star$ *as the solution to the following projection*

$$\min_{\mathbf{x}_j \in \mathbb{R}^p} \quad \sum_j \|\mathbf{x}_j - \hat{x}_j \mathbf{e}\|^2 \quad s.t. \quad \begin{array}{l} \langle \mathbf{e}, \sum_j W_{i,j} \mathbf{x}_j \rangle - \hat{z}_i \leq \rho_i \text{ for all } i, \\ \|\hat{z}_i \mathbf{e} - \sum_j W_{i,j} \mathbf{x}_j / 2\| - \|\sum_j W_{i,j} \mathbf{x}_j / 2\| \leq \rho_i \text{ for all } i. \end{array}$$

*If* $\rho_{\max} \|\mathbf{W}\|^2 \|(\mathbf{WW}^T)^{-1} (\mathbf{W}\hat{\mathbf{x}} - \hat{\mathbf{z}})\|_\infty + \rho_{\max}^2 (1 + \|\mathbf{W}\|^2 \|(\mathbf{WW}^T)^{-1}\|_\infty) < \hat{z}_{\min}^2$ *holds with* $\rho_{\max} = \max_i \rho_i$ *and* $\hat{z}_{\min} = \min_i \hat{z}_i$, *then* $\mathbf{x}_j^\star$ *is unique and satisfies* $\|\mathbf{x}_j^\star\| = |\langle \mathbf{e}, \mathbf{x}_j^\star \rangle|$ *for all* $j$.

We defer the proof of Lemma 6.2 to Appendix D, but note that the main idea is to use the *lossy* S-lemma to solve the minimization of one quadratic (the distance) over several quadratic constraints (the hyperboloids). Theorem 2.2 then follows immediately from Lemma 6.2 and Corollary 2.3.

# 7 Looseness for multiple layers

Unfortunately, the SDP relaxation is not usually tight for more than a single layer. Let $\mathbf{f}(x) = \text{ReLU}(\text{ReLU}(x))$ denote a two-layer neural network with a single neuron per layer. The corresponding instance of problem (B) is essentially the same as problem (2.3) from Section 5 for the one-layer one-neuron network, because $\text{ReLU}(\text{ReLU}(x)) = \text{ReLU}(x)$ holds for all $x$. However, constructing the SDP relaxation and taking the Gram matrix interpretation reveals the following (with $p = 4$)

$$L_{\text{lb}} = \min_{\mathbf{x}, \mathbf{z}, \mathbf{e} \in \mathbb{R}^p} \|\mathbf{x} - \hat{x} \mathbf{e}\| \quad \text{s.t.} \quad \begin{array}{l} \langle \mathbf{z}, \mathbf{e} \rangle \geq \max\{\langle \mathbf{x}, \mathbf{e} \rangle, 0\}, \ \|\mathbf{z}\|^2 \leq \langle \mathbf{z}, \mathbf{x} \rangle, \\ \langle \mathbf{z}, \mathbf{e} \rangle - \hat{z} \leq \rho, \ \|\hat{z} \mathbf{e} - \mathbf{z}/2\| - \|\mathbf{z}/2\| \leq \rho, \end{array} \quad (7.1)$$

which is *almost* the same as problem (5.2) from Section 5, except that the convex ball constraint $\|\hat{z} \mathbf{e} - \mathbf{z}\| \leq \rho$ has been replaced by a nonconvex hyperboloid. As we will see, it is this hyperbolic geometry that makes it harder for the SDP relaxation to be tight.

Denote $x^\star$ as the solution to both instances of problem (B). The point $\mathbf{u} = x^\star \mathbf{e}$ must be the unique solution to (7.1) and (5.2) in order for their respective SDP relaxations to be tight. Now, suppose that $\hat{x} < 0$ and $\hat{z} > \rho > 0$, so that both instances of problem (B) have $x^\star = \hat{z} - \rho > 0$. Both (5.2) and (7.1) are convex over $\mathbf{x}$; fixing $\mathbf{z}$ and optimizing over $\mathbf{x}$ in each case yields $\|\mathbf{x}^\star - \hat{x} \mathbf{e}\| = \|\mathbf{z}\| - \hat{x} \cos\theta$ where $\cos\theta = \langle \mathbf{e}, \mathbf{z} \rangle / \|\mathbf{z}\|$. In order for $\mathbf{u}$ to be the unique solution, we need $\|\mathbf{z}\| - \hat{x} \cos\theta$ to be globally minimized at $\mathbf{z} = \mathbf{u}$. As shown in Figure 4, $\|\mathbf{z}\|$ is clearly minimized at $\mathbf{z}^\star = \mathbf{u}$ over the ball constraint $\|\hat{z} \mathbf{e} - \mathbf{z}\| \leq \rho$, but the same is not obviously true for the hyperbolid $\|\hat{z} \mathbf{e} - \mathbf{z}/2\| - \|\mathbf{z}/2\| \leq \rho$. Some detailed computation readily confirm the geometric intuition that $\mathbf{u}$ is generally a local minimum over the circle, but not over the hyperbola.

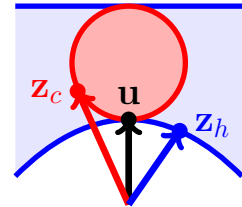

**Fig. 4** – Any point $\mathbf{z}_c$ on the circle clearly satisfies $\|\mathbf{z}_c\| > \|\mathbf{u}\|$, but a point $\mathbf{z}_h$ on the hyperbola may have $\|\mathbf{z}_h\| \approx \|\mathbf{u}\|$.

# 8 Numerical experiments

**Dataset and setup.** We use the MNIST dataset of $28 \times 28$ images of handwritten digits, consisting of 60,000 training images and 10,000 testing images. We remove and set aside the final 1,000 images from the training set as the verification set. All our experiments are performed on an Intel Xeon E3-1230 CPU (4-core, 8-thread, 3.2-3.6 GHz) with 32 GB of RAM.

**Architecture.** We train two small fully-connected neural network ("dense-1" and "dense-3") whose SDP relaxations can be quickly solved using MOSEK [54], and a larger convolutional network

("CNN") whose SDP relaxation must be solved using a custom algorithm described below. The "dense-1" and "dense-3" models respectively have one and three fully-connected layer(s) of 50 neurons, and are trained on a $4 \times 4$ maxpooled version of the training set (each image is downsampled to $7 \times 7$). The "CNN" model has two convolutional layers (stride 2) with 16 and 32 filters (size $4 \times 4$) respectively, followed by a fully-connected layer with 100 neurons, and is trained on the original dataset of $28 \times 28$ images. All models are implemented in tensorflow and trained over 50 epochs using the SGD optimizer (learning rate 0.01, momentum 0.9, "Nesterov" true).

**Rank-2 Burer-Monteiro algorithm ("BM2").** We use a rank-2 Burer–Monteiro algorithm to solve instances of the SDP relaxation on the "CNN" model, by applying a local optimization algorithm to the following (see Appendix F for a detailed derivation and implementation details)

$$\min_{\mathbf{u}_k, \mathbf{v}_k \in \mathbb{R}^n} \quad \|\mathbf{u}_0 - \hat{\mathbf{x}}\|^2 + \|\mathbf{v}_0\|^2 \tag{BM2}$$

$$\text{s.t.} \quad \text{diag}(\mathbf{u}_{k+1}\mathbf{u}_{k+1}^T + \mathbf{v}_{k+1}\mathbf{v}_{k+1}^T) \leq \text{diag}((\mathbf{W}_k\mathbf{u}_k + \mathbf{b}_k)\mathbf{u}_{k+1}^T + \mathbf{W}_k\mathbf{v}_{k+1}\mathbf{v}_{k+1}^T)$$

$$\mathbf{u}_{k+1} \geq \max\{0, \mathbf{W}_k\mathbf{u}_k + \mathbf{b}_k\}, \qquad \|\mathbf{u}_\ell - \hat{\mathbf{z}}\|^2 + \|\mathbf{v}_\ell\|^2 \leq \rho^2 \qquad \text{for all } k.$$

Let $\{\mathbf{u}_k^\star, \mathbf{v}_k^\star\}$ be a locally optimal solution satisfying the first- and second-order optimality conditions (see e.g. [55, Chapter 12]). If $\mathbf{v}_0^\star = 0$, then by induction $\mathbf{u}_{k+1}^\star = \text{ReLU}(\mathbf{W}_k\mathbf{u}_k^\star + \mathbf{b}_k)$ and $\mathbf{v}_k^\star = 0$ for all $k$. It then follows from a well-known result of Burer and Monteiro [56] (see also [57, 58] and in particular [59, Lemma 1]) that $\{\mathbf{u}_k^\star, \mathbf{v}_k^\star\}$ corresponds to a rank-1 solution of the SDP relaxation, and is therefore *globally optimal*. Of course, such a solution must not exist if the relaxation is loose; even when it does exist, the algorithm might still fail to find it if it gets stuck in a *spurious* local minimum with $\mathbf{v}_0^\star \neq 0$. Our experience is that the algorithm consistently succeeds whenever the relaxation is tight, but admittedly this is not guaranteed.

**Tightness for problem (B).** Our theoretical results suggest that the SDP relaxation for problem (B) should be tight for one layer and loose for multiple layers. To verify, we consider the first $k$ layers of the "dense-3" and "CNN" models over a range of radii $\rho$. In each case, we solve 1000 instances of the SDP relaxation, setting $\hat{\mathbf{x}}$ to be a new image from the verification set, and $\hat{\mathbf{z}} = \mathbf{f}(\mathbf{u})$ where $\mathbf{u}$ is the *previous* image used as $\hat{\mathbf{x}}$. MOSEK solved each instance of "dense-3" in 5-20 minutes and BM2 solved each instance of "CNN" in 15-60 minutes. We mark $\mathbf{G}^\star$ as numerically rank-1 if $\lambda_1(\mathbf{G}^\star)/\lambda_2(\mathbf{G}^\star) > 10^3$, and plot the success rates in Figure 5a. Consistent with Theorem 2.2, the relaxation over one layer is most likely to be loose for intermediate values of $\rho$. Consistent with Corollary 2.3, the relaxation eventually becomes tight once $\rho$ is large enough to yield a trivial solution. The results for CNN are dramatic, with an 100% success rate over a single layer, and a 0% success rate for two (and more) layers. BM2 is less successful for very large and very small $\rho$ in part due to numerical issues associated with the factor-of-two exponent in $\|\mathbf{z} - \hat{\mathbf{z}}\|^2 \leq \rho^2$.

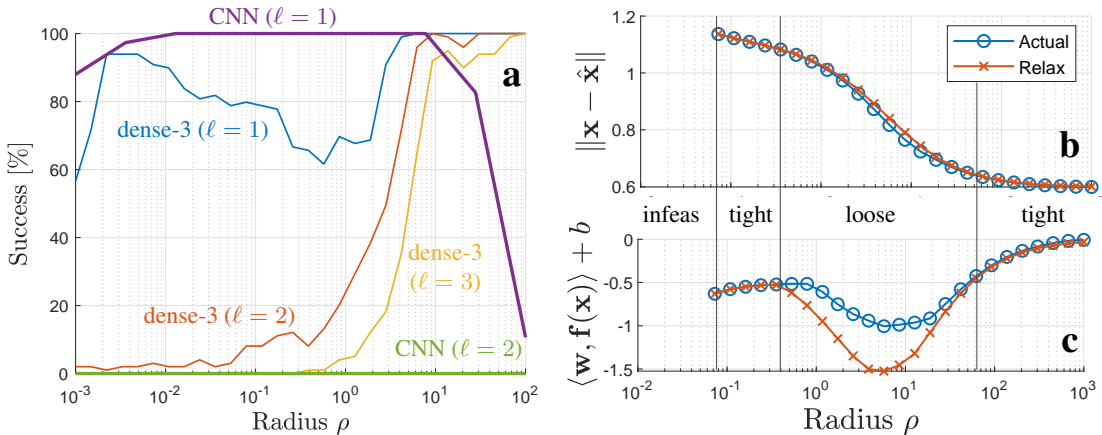

**Fig. 5 – a.** The SDP relaxation for problem (B) is generally tight over a single layer, and loose over multiple layers. **b.** Viewing problem (A) as (B) taken at the limit $\rho \to \infty$, the resulting SDP relaxation can be close to, but not exactly, tight, for finite values of $\rho$. **c.** The SDP relaxation of (B) can produce a near-optimal attack $\mathbf{x}$ satisfying $\langle \mathbf{w}, \mathbf{f}(\mathbf{x}) \rangle + b < 0$ for problem (A), even when relaxation itself is not actually tight.

**Application to problem (A).** Viewing problem (A) as problem (B) in the limit $\rho \to \infty$, we consider a finite range of values for $\rho$, and solve the corresponding SDP relaxation with $\hat{\mathbf{z}} = -\mathbf{w}(b/\|\mathbf{w}\|^2 + \rho/\|\mathbf{w}\|)$. Here, the SDP relaxation is generally loose, so BM2 does not usually succeed, and we must resort to using MOSEK to solve it on the small "dense-1" model. Figure 5b compares the relaxation objective $\sqrt{\text{tr}(\mathbf{X}) - 2\langle\hat{\mathbf{x}}, \mathbf{x}\rangle + \|\hat{\mathbf{x}}\|^2}$ with the actual distance $\|\mathbf{x} - \hat{\mathbf{x}}\|$, while Figure 5c compares the feasibility predicted by the relaxation $\langle\mathbf{w}, \mathbf{z}\rangle + b$ with the actual feasibility $\langle\mathbf{w}, \mathbf{f}(\mathbf{x})\rangle + b$. The relaxation is tight for $0.07 \le \rho \le 0.4$ and $\rho \ge 60$ so the plots coincide. The relaxation is loose for $0.4 \le \rho \le 60$, and the relaxation objective is strictly greater than the actual distance because $\mathbf{X} \succ \mathbf{x}\mathbf{x}^T$. The resulting attack $\mathbf{x}$ must fail to satisfy $\|\mathbf{f}(\mathbf{x}) - \hat{\mathbf{z}}\| \le \rho$, but in this case it is still *always* feasible for problem (A). For $\rho < 0.07$, the SDP relaxation is infeasible, so we deduce that the output target $\hat{\mathbf{z}}$ is not actually feasible.

## 9 Conclusions

This paper presented a geometric study of the SDP relaxation of the ReLU. We split the a modification of the robustness certification problem into the composition of a convex projection onto a spherical cap, and a nonconvex projection onto a hyperboloidal cap, so that the relaxation is tight if and only if the latter projection lies on the major axis of the hyperboloid. This insight allowed us to completely characterize the tightness of the SDP relaxation over a single neuron, and partially characterize the case for the single layer. The multilayer case is usually loose due to the underlying hyperbolic geometry, and this implies looseness in the SDP relaxation of the original certification problem. Our rank-2 Burer-Monteiro algorithm was able to solve the SDP relaxation on a convolution neural network, but better algorithms are still needed before models of realistic scales can be certified.

## Broader Impact

This work contributes towards making neural networks more robust to adversarial examples. This is a crucial roadblock before neural networks can be widely adopted in safety-critical applications like self-driving cars and smart grids. The ultimate, overarching goal is to take the high performance of neural networks—already enjoyed by applications in computer vision and natural language processing—and extend towards applications in societal infrastructure.

Towards this direction, SDP relaxations allow us to make mathematical guarantees on the robustness of a given neural network model. However, a blind reliance on mathematical guarantees leads to a false sense of security. While this work contributes towards robustness of neural networks, much more work is needed to understand the appropriateness of neural networks for societal applications in the first place.

## Acknowledgments

The author is grateful to Salar Fattahi, Cedric Josz, and Yi Ouyang for early discussions and detailed feedback on several versions of the draft. Partial financial support was provided by the National Science Foundation under award ECCS-1808859.

## Footnotes

[1] If a rank-1 solution exists but is nonunique, then we do not consider the SDP relaxation tight because the rank-1 solution cannot usually be found in polynomial time. Indeed, an interior-point method converges onto a maximum rank solution, but this can be rank-1 only if it is unique.

[2]To avoid visual clutter we will abbreviate $\sum_{j=1}^n x_j$ and "for all $i \in \{1, 2, \ldots, n\}$" as $\sum_j x_j$ and "for all $i$" whenever the ranges of indices are clear from context.

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
