[Supplementary Material]

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

# A Uniqueness of a Rank-1 Solution

Consider the rank-constrained semidefinite program

$$\underset{\mathbf{x} \in \mathbb{R}^n, \, \mathbf{X} \in \mathbb{R}^{n \times n}}{\text{minimize}} \qquad \langle \mathbf{D}, \mathbf{X} \rangle + \langle \mathbf{f}, \mathbf{x} \rangle \tag{A.1}$$

$$\text{subject to} \qquad \langle \mathbf{A}_i, \mathbf{X} \rangle + \langle \mathbf{b}_i, \mathbf{x} \rangle \leq c_i \quad \text{for all } i \in \{1, 2, \ldots, m\},$$

$$\begin{bmatrix} 1 & \mathbf{x}^T \\ \mathbf{x} & \mathbf{X} \end{bmatrix} \succeq 0, \quad \text{rank}(\mathbf{X}) \leq p,$$

and its corresponding nonconvex optimization interpretation

$$\underset{\mathbf{v}_1, \mathbf{v}_2, \ldots, \mathbf{v}_n \in \mathbb{R}^p}{\text{minimize}} \qquad \sum_{k=1}^{n} \sum_{j=1}^{n} \langle \mathbf{v}_j, \mathbf{v}_k \rangle \langle \mathbf{D}, \mathbf{e}_k \mathbf{e}_j^T \rangle + \sum_{j=1}^{n} \langle \mathbf{e}, \mathbf{v}_j \rangle \langle \mathbf{f}, \mathbf{e}_j \rangle \tag{A.2}$$

$$\text{subject to} \qquad \sum_{k=1}^{n} \sum_{j=1}^{n} \langle \mathbf{v}_j, \mathbf{v}_k \rangle \langle \mathbf{A}_i, \mathbf{e}_k \mathbf{e}_j^T \rangle + \sum_{j=1}^{n} \langle \mathbf{e}, \mathbf{v}_j \rangle \langle \mathbf{b}_i, \mathbf{e}_j \rangle \leq c_i \quad \text{for all } i \in \{1, 2, \ldots, m\},$$

where $\mathbf{e}$ is an arbitrary, fixed unit vector satisfying $\|\mathbf{e}\| = 1$. Our main result in this section is that we can guarantee a rank-1 solution to (A.1) to be *unique*, and hence computable via an interior-point method, by verifying that every solution to (A.2) is collinear with the unit vector $\mathbf{e}$.

**Definition A.1.** Fix $\mathbf{e} \in \mathbb{R}^p$ with $\|\mathbf{e}\| = 1$. We say that $\mathbf{v} \in \mathbb{R}^p$ is *collinear* or that it satisfies *collinearity* if $|\langle \mathbf{e}, \mathbf{v} \rangle| = \|\mathbf{v}\|$.

**Theorem A.2** (Unique rank-1). *Fix $\mathbf{e} \in \mathbb{R}^p$ with $\|\mathbf{e}\| = 1$, and write $\mathcal{V}^\star$ as the resulting solution set associated with (A.2). Then, problem (A.1) has a unique solution satisfying $\mathbf{X}^\star = \mathbf{x}^\star (\mathbf{x}^\star)^T$ if and only if $\mathbf{v}_1^\star, \mathbf{v}_2^\star, \ldots, \mathbf{v}_n^\star$ are collinear for all $(\mathbf{v}_1^\star, \mathbf{v}_2^\star, \ldots, \mathbf{v}_n^\star) \in \mathcal{V}^\star$.*

We briefly defer the proof of Theorem A.2 to discuss its consequences. In the case of ReLU constraints, if the input vectors $\mathbf{x}_j$ are collinear, then the output vectors $\mathbf{z}_i$ are also collinear.

**Lemma A.3** (Propagation of collinearity). *Fix $\mathbf{e} \in \mathbb{R}^p$ with $\|\mathbf{e}\| = 1$. Under the ReLU constraints $\langle \mathbf{e}, \mathbf{z} \rangle \geq \max\{0, \langle \mathbf{e}, \sum_j w_j \mathbf{x}_j \rangle\}$ and $\langle \mathbf{z}, \mathbf{z} \rangle \leq \langle \mathbf{z}, \sum_j w_j \mathbf{x}_j \rangle$, if $\mathbf{x}_j$ is collinear for all $j$, then $\mathbf{z}$ is also collinear.*

*Proof.* Let $\mathbf{x}_j = x_j \mathbf{e}$ for all $j$, and write $\alpha = \sum_j w_j x_j$ for clarity. Observe that $\langle \mathbf{z}, \mathbf{z} \rangle \leq \langle \mathbf{z}, \sum_j w_j \mathbf{x}_j \rangle = \sum_j w_j x_j \langle \mathbf{z}, \mathbf{e} \rangle = \alpha \langle \mathbf{z}, \mathbf{e} \rangle$. If $\alpha < 0$, then $\langle \mathbf{z}, \mathbf{e} \rangle = 0$ and $\langle \mathbf{z}, \mathbf{e} \rangle = 0$ as claimed. If $\alpha \geq 0$, then $\langle \mathbf{e}, \mathbf{z} \rangle \geq \langle \mathbf{e}, \sum_j w_j \mathbf{x}_j \rangle = \sum_j w_j x_j = \alpha$. Combined with the above, this yields $\langle \mathbf{z}, \mathbf{z} \rangle \leq \alpha \langle \mathbf{z}, \mathbf{e} \rangle \leq \langle \mathbf{z}, \mathbf{e} \rangle^2$. We actually have $\langle \mathbf{z}, \mathbf{z} \rangle = \langle \mathbf{z}, \mathbf{e} \rangle^2$ as claimed, because $\langle \mathbf{z}, \mathbf{z} \rangle \geq \langle \mathbf{z}, \mathbf{e} \rangle^2$ already holds by the Cauchy–Schwarz inequality. $\qquad\square$

Hence, the conditions for the uniquess of the rank-1 solution throughout the main body of the paper are simply special cases of Theorem A.2.

*Proof of Lemma 5.1.* Observe that the semidefinite program (5.1) is a special instance of (A.1), and that its nonconvex interpretation (5.2) is the corresponding instance of (A.2). In the one-neuron case, Lemma A.3 says that if $\mathbf{x}^\star$ satisfies collinearity, then $\mathbf{z}^\star$ also satisfies collinearity. Or put in another way, $\mathbf{x}^\star$ and $\mathbf{z}^\star$ satisfy collinearity if and only if $\mathbf{x}^\star$ satisfies collinearity. Using the latter as an equivalent condition for the former and substituting into Theorem A.2 yields Lemma 5.1 as desired. $\qquad\square$

*Proof of Lemma 6.1.* We repeat the proof of Lemma 5.1, but note that $\mathbf{x}_j^\star$ and $\mathbf{z}_i^\star$ satisfy collinearity for all $i$ and $j$ if and only if $\mathbf{x}_j^\star$ satisfies collinearity for all $j$. Using the latter as an equivalent condition for the former and substituting into Theorem A.2 yields Lemma 6.1 as desired. $\qquad\square$

The main intuition behind the proof of Theorem A.2 is that a non-collinear solution to (A.2) corresponds to a high rank solution to (A.1) with $\text{rank}(\mathbf{X}^\star) > 1$. In turn, a rank-1 solution is unique

if and only if there exists no high-rank solutions; see [60, Theorem 2.4]. To make these ideas rigorous, we begin by reviewing some preliminaries. First, without loss of generality, we can fix $\mathbf{e} = \mathbf{e}_1$, that is, the first canonical basis vector. If we wish to solve (A.2) with a different $\mathbf{e} = \mathbf{e}'$, then we simply need to find an orthonormal matrix $\mathbf{U}$ for which $\mathbf{e}' = \mathbf{U}\mathbf{e}$, for example, using the Gram-Schmidt process. Given a solution $\mathbf{v}_1, \mathbf{v}_2, \ldots, \mathbf{v}_n$ (A.2) with $\mathbf{e} = \mathbf{e}_1$, setting $\mathbf{v}'_j = \mathbf{U}\mathbf{v}_j$ yields a solution $\mathbf{v}'_1, \mathbf{v}'_2, \ldots, \mathbf{v}'_n$ to (A.2) with $\mathbf{e} = \mathbf{e}'$, because $\langle \mathbf{v}_k, \mathbf{v}_j \rangle = \langle \mathbf{U}\mathbf{v}_k, \mathbf{U}\mathbf{v}_j \rangle = \langle \mathbf{v}'_k, \mathbf{v}'_j \rangle$ and $\langle \mathbf{e}_1, \mathbf{v}_j \rangle = \langle \mathbf{U}\mathbf{e}_1, \mathbf{U}\mathbf{v}_j \rangle = \langle \mathbf{e}', \mathbf{v}'_j \rangle$.

The equivalence between (A.1) and (A.2) is established by using the solution to one problem to construct a *corresponding solution* satisfying the following relationship

$$\langle \mathbf{x}, \mathbf{e}_j \rangle = \langle \mathbf{e}, \mathbf{v}_j \rangle, \qquad \langle \mathbf{X}, \mathbf{e}_j \mathbf{e}_k^T \rangle = \langle \mathbf{v}_j, \mathbf{v}_k \rangle,$$

for the other problem. In one direction, given a solution $\mathbf{v}_1, \mathbf{v}_2, \ldots, \mathbf{v}_n \in \mathbb{R}^p$ to (A.2), the corresponding solution to (A.1) is simply

$$\mathbf{x} = [\langle \mathbf{e}, \mathbf{v}_j \rangle]_{j=1}^n = \mathbf{V}^T \mathbf{e}, \qquad \mathbf{X} = [\langle \mathbf{v}_j, \mathbf{v}_k \rangle]_{j,k=1}^n = \mathbf{V}^T \mathbf{V},$$

where $\mathbf{V} = \begin{bmatrix} \mathbf{v}_1 & \mathbf{v}_2 & \cdots & \mathbf{v}_n \end{bmatrix} \in \mathbb{R}^{p \times n}$. In the other direction, given a solution $\mathbf{x}$ and $\mathbf{X}$ to (A.1), we factorize $\mathbf{X} - \mathbf{x}\mathbf{x}^T = \tilde{\mathbf{V}}^T \tilde{\mathbf{V}}$ so that

$$\begin{bmatrix} 1 & \mathbf{x}^T \\ \mathbf{x} & \mathbf{X} \end{bmatrix} = [\mathbf{e}_1 \quad \mathbf{V}]^T [\mathbf{e}_1 \quad \mathbf{V}], \quad \mathbf{V} = \begin{bmatrix} \mathbf{x}^T \\ \tilde{\mathbf{V}} \end{bmatrix} = [\mathbf{v}_1 \quad \mathbf{v}_2 \quad \cdots \quad \mathbf{v}_n] \in \mathbb{R}^{p \times n}.$$

Then, $\mathbf{v}_1, \mathbf{v}_2, \ldots, \mathbf{v}_n$ is a corresponding solution to (A.2) with $\mathbf{e} = \mathbf{e}_1$.

*Proof of Theorem A.2.* ($\Rightarrow$) Given a rank-1 solution $\mathbf{X}^\star = \mathbf{x}^\star (\mathbf{x}^\star)^T$ of the relaxation (A.1), we set $x_j^\star = \langle \mathbf{e}_j, \mathbf{x}^\star \rangle$ and $\mathbf{v}_j^\star = \langle \mathbf{e}_j, \mathbf{x}^\star \rangle \mathbf{e}$ to obtain a corresponding solution $\mathbf{v}_1^\star, \mathbf{v}_2^\star, \ldots, \mathbf{v}_n^\star$ to (A.2) that satisfies collinearity. By contradiction, suppose that there exists another solution $\mathbf{v}'_1, \mathbf{v}'_2, \ldots, \mathbf{v}'_n$ to (A.2) that does not satisfy collinearity, meaning that there exists some $s$ such that $|\langle \mathbf{e}, \mathbf{v}'_s \rangle| \neq \|\mathbf{v}'_s\|$. Then, its corresponding solution $\mathbf{x}', \mathbf{X}'$ is distinct from $\mathbf{x}^\star, \mathbf{X}^\star$, because $|\langle \mathbf{e}, \mathbf{v}'_s \rangle| \neq \|\mathbf{v}'_s\|$ but $|\langle \mathbf{e}, \mathbf{v}_s^\star \rangle| = \|\mathbf{v}_s^\star\|$, so we can have either $\langle \mathbf{X}^\star - \mathbf{X}', \mathbf{e}_s \mathbf{e}_s^T \rangle = \|\mathbf{v}_s^\star\|^2 - \|\mathbf{v}'_s\|^2 = 0$ or $\langle \mathbf{x}^\star - \mathbf{x}', \mathbf{e}_s \rangle = \langle \mathbf{e}, \mathbf{v}_s^\star - \mathbf{v}'_s \rangle = 0$ but not both at the same time. This contradicts the hypothesis that $\mathbf{X}^\star$ is a unique solution.

($\Leftarrow$) Without loss of generality, we assume that $\mathbf{e} = \mathbf{e}_1$. Given a solution $\mathbf{v}_1^\star, \mathbf{v}_2^\star, \ldots, \mathbf{v}_n^\star$ to (A.2) satisfying collinearity, we set $x_j^\star = \langle \mathbf{e}, \mathbf{v}_j^\star \rangle$, $\mathbf{x}^\star = [x_j^\star]_{j=1}^n$, and $\mathbf{X}^\star = \mathbf{x}^\star (\mathbf{x}^\star)^T$, in order to obtain a corresponding rank-1 solution to (A.1). By contradiction, suppose that there exists another solution $\mathbf{x}', \mathbf{X}'$ to (A.1) that is distinct from $\mathbf{x}^\star, \mathbf{X}^\star$, with corresponding solution $\mathbf{v}'_1, \mathbf{v}'_2, \ldots, \mathbf{v}'_n$ to (A.2). This solution $\mathbf{v}'_1, \mathbf{v}'_2, \ldots, \mathbf{v}'_n$ must satisfy collinearity, or else our hypothesis is immediately violated. Under collinearity, we again set $x'_j = \langle \mathbf{e}, \mathbf{v}'_j \rangle$ such that $\mathbf{x}' = [x'_j]_{j=1}^n$ and $\mathbf{X}' = \mathbf{x}'(\mathbf{x}')^T$. Then, the following

$$\mathbf{v}_j = \frac{1}{2}\mathbf{e}_1(x_j^\star + x'_j) + \frac{1}{2}\mathbf{e}_2(x_j^\star - x'_j)$$

yields another solution, since

$$\langle \mathbf{e}_1, \mathbf{v}_j \rangle = \frac{1}{2}(x_j^\star + x'_j) = \frac{1}{2}(\langle \mathbf{e}_1, \mathbf{v}_j^\star \rangle + \langle \mathbf{e}_1, \mathbf{v}'_j \rangle)$$

$$\langle \mathbf{v}_j, \mathbf{v}_k \rangle = \frac{1}{4}(x_j^\star + x'_j)(x_k^\star + x'_k) + \frac{1}{4}(x_j^\star - x'_j)(x_k^\star - x'_k) = \frac{1}{2}(x_j^\star x_k^\star + x'_j x'_k)$$

$$= \frac{1}{2}(\langle \mathbf{v}_j^\star, \mathbf{v}_k^\star \rangle + \langle \mathbf{v}'_j, \mathbf{v}'_k \rangle)$$

In order for $\mathbf{x}', \mathbf{X}'$ is distinct from $\mathbf{x}^\star, \mathbf{X}^\star$, there must be some choice of $s$ such that $x_s^\star \neq x'_s$, but this means that $\mathbf{v}_s$ does not satisfy collinearity, since $\langle \mathbf{e}_2, \mathbf{v}_s \rangle = \frac{1}{2}(x_s^\star - x'_s) \neq 0$. This contradicts the hypothesis that all solutions $\mathbf{v}_1, \mathbf{v}_2, \ldots, \mathbf{v}_n$ to (A.2) satisfy collinearity. $\square$

# B  Projection onto ReLU Feasbility Set

Fix $\mathbf{e}, \mathbf{x} \in \mathbb{R}^p$ and $\hat{z} \in \mathbb{R}$. Let $\alpha = \max\{\langle \mathbf{e}, \mathbf{x} \rangle, 0\}$, and define $\phi$ as the projection distance onto the spherical cap defined by the "ReLU feasible set" (5.4), restated here as

$$\phi = \min_{\mathbf{z} \in \mathbb{R}^p} \quad \|\mathbf{z} - \hat{z}\mathbf{e}\| \quad \text{s.t.} \quad \langle \mathbf{e}, \mathbf{z} \rangle \geq \alpha, \quad \|\mathbf{z}\|^2 \leq \langle \mathbf{z}, \mathbf{x} \rangle. \tag{B.1}$$

In the main text, we used intuitive, geometric arguments to prove that

$$\phi = \begin{cases} \alpha - \hat{z} & \hat{z} \le \alpha, \\ \|\hat{z}\mathbf{e} - \mathbf{x}/2\| - \|\mathbf{x}/2\| & \hat{z} > \alpha. \end{cases} \tag{B.2}$$

In this section, we will rigorously verify (B.2) and then prove that the conditional statements are unnecessary, in that $\phi$ simply takes on the larger of the two values, as in

$$\phi = \max\{\alpha - \hat{z}, \quad \|\hat{z}\mathbf{e} - \mathbf{x}/2\| - \|\mathbf{x}/2\|\}. \tag{B.3}$$

This was stated in the main text as Lemma 5.2.

We first rigorously verify (B.2) by: 1) relaxing a constraint for a specified case; 2) solving the relaxation in closed-form; 3) verifying that the closed-form solution satisfies the original constraints, and must therefore be optimal for the original problem. In the case of $\hat{z} \le \alpha$, the following relaxation

$$\phi_{\mathrm{lb1}} = \min_{\mathbf{z}\in\mathbb{R}^p}\{\|\mathbf{z} - \hat{z}\mathbf{e}\| \quad : \quad \langle\mathbf{e}, \mathbf{z}\rangle \ge \alpha\}$$

has solution $\mathbf{z}^\star = \alpha\mathbf{e}$ that is clearly feasible for (B.1) since $\|\mathbf{z}^\star\|^2 = \langle\mathbf{z}^\star, \mathbf{x}\rangle = \alpha^2$. Hence, this $\mathbf{z}^\star$ must be optimal; its objective $\|\mathbf{z}^\star - \hat{z}\mathbf{e}\| = \alpha - \hat{z}$ yields the desired value of $\phi$.

In the case of $\hat{z} > \alpha$, the following relaxation

$$\phi_{\mathrm{lb2}} = \min_{\mathbf{z}\in\mathbb{R}^p}\{\|\mathbf{z} - \hat{z}\mathbf{e}\|^2 \quad : \quad \|\mathbf{z}\|^2 \le \langle\mathbf{z}, \mathbf{x}\rangle\},$$

must have an active constraint at optimality. Otherwise, the solution would be $\mathbf{z} = \hat{z}\mathbf{e}$, but this cannot be feasible as $\hat{z}^2 = \|\mathbf{z}\| \le \langle\mathbf{z}, \mathbf{x}\rangle = \hat{z}\langle\mathbf{e}, \mathbf{x}\rangle \le \hat{z}\alpha$ would contradict $\hat{z} > \alpha \ge 0$. Applying Lagrange multipliers, the solution reads $\mathbf{z}^\star = t \cdot \hat{z}\mathbf{e} + (1 - t) \cdot \mathbf{x}/2$ where $t = \|\mathbf{x}/2\|/\|\hat{z}\mathbf{e} - \mathbf{x}/2\|$ is chosen to make the constraint active. We will need the following lemma to verify that $\langle\mathbf{e}, \mathbf{z}^\star\rangle \ge \alpha$.

**Lemma B.1.** *Let $|v| \le R$. If $u > \sqrt{R^2 - v^2}$, then $Ru/\sqrt{u^2 + v^2} \ge \sqrt{R^2 - v^2}$.*

*Proof.* We will prove that if $u^2 + v^2 > R^2$ then $R^2u^2/(u^2 + v^2) + v^2 \ge R^2$. By contradiction, suppose that $R^2u^2/(u^2 + v^2) + v^2 < R$. If $u^2 + v^2 = 0$, then the premise is already false. Otherwise, we multiply by $u^2 + v^2 > 0$ to yield $R^2u^2 + v^2(u^2 + v^2) < R^2(u^2 + v^2)$, or equivalently $v^2(u^2 + v^2 - R^2) < 0$. This last condition is only possible if $v \ne 0$ and $u^2 + v^2 < R^2$, but this again contradicts the premise. $\square$

For $u = 2\hat{z} - \langle\mathbf{e}, \mathbf{x}\rangle$, $v = \sqrt{\|\mathbf{x}\|^2 - \langle\mathbf{e}, \mathbf{x}\rangle^2}$, and $R = \|\mathbf{x}\|$, observe that

$$t = \frac{\|\mathbf{x}/2\|}{\|\hat{z}\mathbf{e} - \mathbf{x}/2\|} = \frac{R}{\sqrt{u^2 + v^2}}, \qquad \alpha = \max\{\langle\mathbf{e}, \mathbf{x}\rangle, 0\} = \frac{\langle\mathbf{e}, \mathbf{x}\rangle}{2} + \frac{|\langle\mathbf{e}, \mathbf{x}\rangle|}{2}.$$

Then, $\mathbf{z}^\star = t \cdot \hat{z}\mathbf{e} + (1 - t) \cdot \mathbf{x}/2$ is feasible for (B.1), because substituting $u, v, R$ into Lemma B.1 yields

$$\hat{z} > \alpha \quad \Longleftrightarrow \quad \hat{z} - \frac{\langle\mathbf{e}, \mathbf{x}\rangle}{2} > \frac{|\langle\mathbf{e}, \mathbf{x}\rangle|}{2} \quad \Longrightarrow \quad t \cdot \left(\hat{z} - \frac{\langle\mathbf{e}, \mathbf{x}\rangle}{2}\right) \ge \frac{|\langle\mathbf{e}, \mathbf{x}\rangle|}{2},$$

and this in turn implies that

$$\langle\mathbf{e}, \mathbf{z}^\star\rangle = \frac{\langle\mathbf{e}, \mathbf{x}\rangle}{2} + t \cdot \left(\hat{z} - \frac{\langle\mathbf{e}, \mathbf{x}\rangle}{2}\right) \ge \frac{\langle\mathbf{e}, \mathbf{x}\rangle}{2} + \frac{|\langle\mathbf{e}, \mathbf{x}\rangle|}{2} = \alpha.$$

Hence, this $\mathbf{z}^\star$ must be optimal; its objective $\|\mathbf{z}^\star - \hat{z}\mathbf{e}\| = (1 - t)\|\hat{z}\mathbf{e} - \mathbf{x}/2\|$ yields the desired value of $\phi$.

Finally, we prove (5.5) by showing that the conditional statements in (B.2) are unnecessary.

*Proof of Lemma 5.2.* If $\hat{z} > \alpha$, then clearly $\phi = \phi_{\mathrm{lb2}} \ge 0$ by construction, but $\alpha - \hat{z} < 0$, so $\phi = \max\{\alpha - \hat{z}, \phi_{\mathrm{lb2}}\}$ as desired. For $\hat{z} \le \alpha$, we will proceed by examining two cases. First, suppose that $\hat{z} \ge 0$ and hence $\alpha = \langle\mathbf{e}, \mathbf{x}\rangle$ and $\hat{z} \le \langle\mathbf{e}, \mathbf{x}\rangle$. Then, $\|\hat{z}\mathbf{e} - \mathbf{x}/2\|^2 - \|\mathbf{x}/2\|^2 = \hat{z}(\hat{z} - \langle\mathbf{e}, \mathbf{x}\rangle) \le 0$, and $\|\hat{z}\mathbf{e} - \mathbf{x}/2\| - \|\mathbf{x}/2\| \le 0$, so $\phi = \max\{\phi_{\mathrm{lb1}}, \|\hat{z}\mathbf{e} - \mathbf{x}/2\| - \|\mathbf{x}/2\|\}$ as desired. In the case

of $\hat{z} \leq 0$, Lemma C.1 shows that $\langle \mathbf{e}, \mathbf{x} \rangle - \hat{z} \leq \rho$ implies $\|\hat{z}\mathbf{e} - \mathbf{x}/2\| - \|\mathbf{x}/2\| \leq \rho$, since with $u = \langle \mathbf{e}, \mathbf{x} \rangle$, $v = \sqrt{\|\mathbf{x}\|^2 - \langle \mathbf{e}, \mathbf{x} \rangle^2}$, $c = |\hat{z}|$, and $a = \rho$, we have

$$\|\mathbf{x}/2 + |\hat{z}|\mathbf{e}\| - \|\mathbf{x}/2\| \leq \rho \quad \Longleftrightarrow \quad \frac{\langle \mathbf{e}, \mathbf{x} \rangle + |\hat{z}|}{\rho} \leq \sqrt{1 + \frac{\|\mathbf{x}\|^2 - \langle \mathbf{e}, \mathbf{x} \rangle^2}{\hat{z}^2 - \rho^2}}$$

but $\langle \mathbf{e}, \mathbf{x} \rangle - \hat{z} \leq \rho$ already implies $\frac{1}{\rho}[\langle \mathbf{e}, \mathbf{x} \rangle + |\hat{z}|] \leq 1$. In particular, the fact that $\langle \mathbf{e}, \mathbf{x} \rangle - \hat{z} \leq \phi_{\text{lb1}}$ implies $\|\hat{z}\mathbf{e} - \mathbf{x}/2\| - \|\mathbf{x}/2\| \leq \phi_{\text{lb1}}$ shows that we have $\phi = \max\{\phi_{\text{lb1}}, \|\hat{z}\mathbf{e} - \mathbf{x}/2\| - \|\mathbf{x}/2\|\}$. $\square$

# C  Projection onto a hyperbola

Fix $\mathbf{e}, \mathbf{x} \in \mathbb{R}^p$ and $\hat{x}, \hat{z}, \rho \in \mathbb{R}$ such that $\hat{z} > \rho > 0$. Define $\psi$ as the projection distance onto the hyperboloidal cap (5.4), restated here

$$\psi = \min_{\mathbf{x} \in \mathbb{R}^p} \|\mathbf{x} - \hat{x}\mathbf{e}\| \quad \text{s.t.} \quad \langle \mathbf{e}, \mathbf{x} \rangle - \hat{z} \leq \rho, \quad \|2\hat{z}\mathbf{e} - \mathbf{x}\| - \|\mathbf{x}\| \leq 2\rho. \tag{C.1}$$

Without loss of generality, we can fix $\mathbf{e} = \mathbf{e}_1$ (see Appendix A), and split the coordinates of $\mathbf{x}$ as in $u = \mathbf{x}[1]$ and $\mathbf{v}[j] = \mathbf{x}[1+j]$ for $j \in \{1, 2, \ldots, p-1\}$ to rewrite (C.1) as the following

$$\psi^2 = \min_{(u,\mathbf{v}) \in \mathbb{R}^p} (u - \hat{x})^2 + \|\mathbf{v}\|^2 \tag{C.2}$$

$$\text{s.t.} \quad u - \hat{z} \leq \rho, \quad \sqrt{(u - 2\hat{z})^2 + \|\mathbf{v}\|^2} - \sqrt{u^2 + \|\mathbf{v}\|^2} \leq 2\rho.$$

Observe that the variable $\mathbf{v} \in \mathbb{R}^{p-1}$ only appears in (C.2) via its norm $\|\mathbf{v}\|$. Hence, (C.2) is equivalent to the following problem

$$\psi^2 = \min_{u,v \in \mathbb{R}} (u - \hat{x})^2 + v^2 \tag{C.3}$$

$$\text{s.t.} \quad u - \hat{z} \leq \rho, \quad \sqrt{(u - 2\hat{z})^2 + v^2} - \sqrt{u^2 + v^2} \leq 2\rho,$$

and a solution $\mathbf{v}^\star$ to (C.2) can be recovered from a solution $v^\star$ to (C.3) by picking any unit vector $\mathbf{s} \in \mathbb{R}^{p-1}$ with $\|\mathbf{s}\| = 1$ and setting $\mathbf{v}^\star = v^\star \mathbf{s}$. We have reduced the projection over a hyperboloid (C.1) into a projection onto a hyperbola (C.3) by taking a quotient over the minor-axis directions. To proceed, we will need the following technical lemma, which is mechanically derived by completing the square and collecting terms.

**Lemma C.1.** *Given semi-major axis $a > 0$, semi-minor axis $b > 0$, and focus $c = \sqrt{a^2 + b^2}$, the following hold*

$$\sqrt{(u - 2c)^2 + v^2} - \sqrt{u^2 + v^2} \leq 2a \quad \Longleftrightarrow \quad \frac{u - c}{a} \geq \sqrt{1 + \frac{v^2}{b^2}}, \tag{C.4a}$$

$$\sqrt{(u + 2c)^2 + v^2} - \sqrt{u^2 + v^2} \leq 2a \quad \Longleftrightarrow \quad \frac{u + c}{a} \leq \sqrt{1 + \frac{v^2}{b^2}}. \tag{C.4b}$$

We use Lemma C.1 to rewrite the hyperbolic constraint in (C.3) in quadratic form, as in

$$\psi^2 = \min_{u,v \in \mathbb{R}} (u - \hat{x})^2 + v^2 \quad \text{s.t.} \quad \frac{u - \hat{z}}{\rho} \leq 1, \quad \frac{(u - \hat{z})^2}{\rho^2} - \frac{v^2}{\hat{z}^2 - \rho^2} \leq 1. \tag{C.5}$$

We will need the following to solve (C.5). This is the main result of this section.

**Theorem C.2** (Axial projection onto a hyperbola). *The problem data $\mathbf{a}, \mathbf{x} \in \mathbb{R}^m$, $\mathbf{c} \in \mathbb{R}^m$ and $b \in \mathbb{R}$ satisfy*

$$\mathbf{a}, \mathbf{c} \neq 0, \quad |\langle \mathbf{a}, \mathbf{x} \rangle - b| - 1 < \|\mathbf{a}\|^2 / \|\mathbf{c}\|^2$$

*if and only if the following projection*

$$(\mathbf{u}^\star, \mathbf{v}^\star) = \arg\min_{\mathbf{u},\mathbf{v}} \left\{ \|\mathbf{u} - \mathbf{x}\|^2 + \|\mathbf{v}\|^2 : (\langle \mathbf{a}, \mathbf{u} \rangle - b)^2 - \langle \mathbf{c}, \mathbf{v} \rangle^2 \leq 1 \right\}$$

*has a unique solution*

$$\mathbf{u}^\star = \mathbf{x} - \mathbf{a} \frac{(\langle \mathbf{a}, \mathbf{x} \rangle - b)}{\|\mathbf{a}\|^2} \left( 1 - \frac{1}{|\langle \mathbf{a}, \mathbf{x} \rangle - b|} \right),$$

$$\mathbf{v}^\star = 0.$$

The proof of Theorem C.2 will span the remainder of this section. Lemma 5.3 is clearly a special instance as applied to (C.5).

*Proof of Lemma 5.3.* If $\hat{x} \geq \hat{z} - \rho$, then relaxing the hyperbolic constraint in (C.5) yields a unique solution of $u^\star = \min\{\hat{x}, \hat{z} + \rho\}$ and $v^\star = 0$. Indeed, this solution also satisfies the hyperbolic constraint, and is therefore optimal for (C.5). Otherwise, if $\hat{x} < \hat{z} - \rho$, then we will use relax the linear constraint in (C.5) and apply Theorem C.2. Here, $\mathbf{a} = 1/\rho$, $b = \hat{z}/\rho$, $\mathbf{c} = 1/\sqrt{\hat{z}^2 - \rho^2}$, and $\mathbf{x} = \hat{x}$, and the condition for (C.5) to have a unique condition $u^\star$ and $v^\star$ with $v^\star = 0$ is

$$|\hat{x} - \hat{z}|/\rho - 1 < (\hat{z}^2 - \rho^2)/\rho^2 \qquad \Longleftrightarrow \qquad |\hat{x} - \hat{z}| < \hat{z}^2/\rho. \tag{C.6}$$

It is easy to verify that the resulting solution is feasible for (C.5), and hence optimal. Under the premise $\hat{x} - \hat{z} < -\rho < 0$, the condition (C.6) is just $\hat{x} > \hat{z} - \hat{z}^2/\rho$, which also implies $\hat{x} \geq \hat{z} - \rho$ because $\hat{z} > \rho$. Hence, we have covered both cases; the condition $(\hat{z} - \hat{x}) < \hat{z}^2/\rho$ guarantees a unique $u^\star$ and $v^\star = 0$ as claimed. $\qquad\square$

We will now prove Theorem C.2. The Euclidean projection onto a hyperbola is the minimization of one quadratic function subject to another quadratic function. This is well-known to be a tractable problem via the S-procedure (see e.g. [52, p. 655] or [53]). In its original form, it states that for two quadratics $f(\mathbf{x})$ and $g(\mathbf{x})$ for which there exists $\mathbf{x}_0$ satisfying $g(\mathbf{x}_0) < 0$, that

$$f(\mathbf{x}) \geq 0 \quad \text{holds for all } \mathbf{x} \text{ satisfying } g(\mathbf{x}) \leq 0$$

if and only if there exists $\lambda \geq 0$ such that

$$f(\mathbf{x}) + \lambda g(\mathbf{x}) \geq 0 \quad \text{holds for all } \mathbf{x}.$$

Clearly, a corollary of the S-procedure is strong duality, as in

$$\min_{\mathbf{x}}\{f(\mathbf{x}) : g(\mathbf{x}) \leq 0) \quad = \quad \max_{\lambda \geq 0} \min_{\mathbf{x}}\{f(\mathbf{x}) + \lambda g(\mathbf{x})\},$$

and so the Karush–Kuhn–Tucker conditions allow us to solve the primal by solving the dual, assuming the existence of a strictly feasible point $\mathbf{x}_0$ with $g(\mathbf{x}_0) < 0$. To proceed, we will need the following technical lemma, which is mechanically derived by applying the Sherman-Morrison identity.

**Lemma C.3** (Rank-1 update). *Given $\mathbf{a}, \mathbf{x} \in \mathbb{R}^m$, $b \in \mathbb{R}$, and $\lambda > -1/\|\mathbf{a}\|^2$, the following projection*

$$\mathbf{u}^\star = \arg\min_{u \in \mathbb{R}^n}\{\|\mathbf{u} - \mathbf{x}\|^2 + \lambda(\langle\mathbf{a}, \mathbf{u}\rangle - b)^2\}$$

*has a unique solution $\mathbf{u}^\star$ satisfying*

$$\mathbf{u}^\star = \mathbf{x} - \lambda\mathbf{a}\left(\frac{\langle\mathbf{a}, \mathbf{x}\rangle - b}{1 + \lambda\|\mathbf{a}\|^2}\right), \qquad \langle\mathbf{a}, \mathbf{u}^\star\rangle - b = \frac{\langle\mathbf{a}, \mathbf{x}\rangle - b}{1 + \lambda\|\mathbf{a}\|^2}.$$

$$\|\mathbf{u}^\star - \mathbf{x}\|^2 + \lambda(\langle\mathbf{a}, \mathbf{u}^\star\rangle - b)^2 = \frac{\lambda(\langle\mathbf{a}, \mathbf{x}\rangle - b)^2}{1 + \lambda\|\mathbf{a}\|^2}$$

We will actually solve the most general form of the projection problem.

**Lemma C.4** (General projection onto a single hyperbola). *Let $\mathbf{a}, \mathbf{x} \in \mathbb{R}^m$, $\mathbf{c}, \mathbf{y} \in \mathbb{R}^m$ and $b, d \in \mathbb{R}$ satisfy $\mathbf{a}, \mathbf{c} \neq 0$. Let $\mathbf{u}^\star \in \mathbb{R}^m$, $\mathbf{v}^\star \in \mathbb{R}^m$ be solutions to the projection*

$$\phi = \min_{\mathbf{u}, \mathbf{v}}\left\{\|\mathbf{u} - \mathbf{x}\|^2 + \|\mathbf{v} - \mathbf{y}\|^2 : (\langle\mathbf{a}, \mathbf{u}\rangle - b)^2 - (\langle\mathbf{c}, \mathbf{v}\rangle - d)^2 \leq 1\right\},$$

*and let $\lambda^\star$ be the unique solution to the Lagrangian dual*

$$\phi_{\text{lb}} = \max_{0 \leq \lambda \leq 1/\|\mathbf{c}\|^2}\left\{\lambda\left[\frac{(\langle\mathbf{a}, \mathbf{x}\rangle - b)^2}{1 + \lambda\|\mathbf{a}\|^2} - \frac{(\langle\mathbf{c}, \mathbf{y}\rangle - d)^2}{1 - \lambda\|\mathbf{c}\|^2} - 1\right]\right\}.$$

*Then, $\phi = \phi_{\text{lb}}$. Moreover the primal solutions are unique if and only if $\lambda^\star < 1/\|\mathbf{c}\|^2$, with values*

$$\mathbf{u}^\star = \mathbf{x} - \lambda^\star\mathbf{a}\left(\frac{\langle\mathbf{a}, \mathbf{x}\rangle - b}{1 + \lambda^\star\|\mathbf{a}\|^2}\right), \qquad \mathbf{v}^\star = y + \lambda^\star\mathbf{c}\left(\frac{\langle\mathbf{c}, \mathbf{y}\rangle - d}{1 - \lambda^\star\|\mathbf{c}\|^2}\right).$$

*Proof.* We define the following two quadratics and corresponding Lagrangian

$$f(\mathbf{u}, \mathbf{v}) = \|\mathbf{u} - \mathbf{x}\|^2 + \|\mathbf{v} - \mathbf{y}\|^2,$$
$$g(\mathbf{u}, \mathbf{v}) = (\langle \mathbf{a}, \mathbf{u} \rangle - b)^2 - (\langle \mathbf{c}, \mathbf{v} \rangle - d)^2 - 1,$$
$$L(\mathbf{u}, \mathbf{v}, \lambda) = f(\mathbf{u}, \mathbf{v}) + \lambda g(\mathbf{u}, \mathbf{v}).$$

Note that $\mathbf{u}_0 = b\mathbf{a}/\|\mathbf{a}\|$ and $\mathbf{v}_0 = d\mathbf{c}/\|\mathbf{c}\|$ satisfies $g(\mathbf{u}_0, \mathbf{v}_0) < 0$, so strong duality holds via the S-procedure. Next, we apply Lemma C.3 to yield the Lagrangian dual $\phi_{\text{lb}}$ via

$$\min_{\mathbf{u}, \mathbf{v}} L(\mathbf{u}, \mathbf{v}, \lambda) = \begin{cases} \lambda \left[ \frac{(\langle \mathbf{a}, \mathbf{x} \rangle - b)^2}{1 + \lambda \|\mathbf{a}\|^2} - \frac{(\langle \mathbf{c}, \mathbf{y} \rangle - d)^2}{1 - \lambda \|\mathbf{c}\|^2} - 1 \right] & \lambda \leq 1/\|\mathbf{c}\|^2, \\ -\infty & \lambda > 1/\|\mathbf{c}\|^2. \end{cases}$$

It is easy to verify that the dual function above is strongly concave over $\lambda$, so the solution $\lambda^\star$ is unique. Finally, if $\lambda^\star < 1/\|\mathbf{c}\|^2$, then the Lagrangian $L(\mathbf{u}, \mathbf{v}, \lambda^\star)$ is strongly convex, and the primal solutions $\mathbf{u}^\star$ and $\mathbf{v}^\star$ are both uniquely determined by minimizing $L(\mathbf{u}, \mathbf{v}, \lambda^\star)$. Otherwise, if $\lambda^\star = 1/\|\mathbf{c}\|^2$, then $L(\mathbf{u}, \mathbf{v}, \lambda^\star)$ is weakly convex over $\mathbf{v}$. Here, $\mathbf{u}^\star$ is uniquely determined by minimizing $L(\mathbf{u}, \mathbf{v}, \lambda^\star)$, but $\mathbf{v}^\star$ can be any choice that satisfies primal feasibility $(\langle \mathbf{a}, \mathbf{u}^\star \rangle - b)^2 - (\langle \mathbf{c}, \mathbf{v}^\star \rangle - d)^2 = 1$, and is therefore nonunique. $\qquad\square$

Finally, we prove Theorem C.2 using Lemma C.4.

*Proof of Theorem C.2.* The axial projection problem of Theorem C.2 is an instance of the more general projection problem in Lemma C.4 with $\mathbf{y} = 0$ and $d = 0$. The intended claim holds so long as $\lambda^\star < 1/\|\mathbf{c}\|^2$. Now, first order optimality in the Lagrangian dual reads

$$\frac{(\langle \mathbf{a}, \mathbf{x} \rangle - b)^2}{(1 + \lambda^\star \|\mathbf{a}\|^2)^2} - \frac{(\langle \mathbf{c}, \mathbf{y} \rangle - d)^2}{(1 - \lambda^\star \|\mathbf{c}\|^2)^2} - 1 = 0,$$

and this implies $1 + \lambda^\star \|\mathbf{a}\|^2 = |\langle \mathbf{a}, \mathbf{x} \rangle - b|$ and hence $\lambda^\star = (|\langle \mathbf{a}, \mathbf{x} \rangle - b| - 1)/\|\mathbf{a}\|^2$. Finally, imposing the bound $\lambda^\star < 1/\|\mathbf{c}\|^2$ on this value yields our desired claim. $\qquad\square$

# D   Projection onto several hyperbolas

Given $\mathbf{W} = [W_{i,j}] \in \mathbb{R}^{m \times n}$, $\hat{\mathbf{x}} = [\hat{x}_j] \in \mathbb{R}^n$, $\hat{\mathbf{z}} = [\hat{z}_i] \in \mathbb{R}^m$, $\mathbf{e} \in \mathbb{R}^p$, and $\rho_i$ satisfying $\hat{z}_i > \rho > 0$, we will partially solve

$$\min_{\mathbf{x}_j \in \mathbb{R}^p} \quad \sum_j \|\mathbf{x}_j - \hat{x}_j \mathbf{e}\|^2 \quad \text{s.t.} \quad \begin{array}{l} \langle \mathbf{e}, \sum_j W_{i,j} \mathbf{x}_j \rangle - \hat{z}_i \leq \rho_i \text{ for all } i, \\ \|\hat{z}_i \mathbf{e} - \sum_j W_{i,j} \mathbf{x}_j / 2\| - \|\sum_j W_{i,j} \mathbf{x}_j / 2\| \leq \rho_i \text{ for all } i. \end{array} \quad \text{(D.1)}$$

Without loss of generality, we can fix $\mathbf{e} = \mathbf{e}_1$ and split the coordinates of $\mathbf{x}_j$ as in $\mathbf{u}[j] = \mathbf{x}_j[1]$ for all $j$ and $\mathbf{v}_k[j] = \mathbf{x}_j[1 + k]$ for all $j, k$ to rewrite (D.1) as the following

$$\min_{\mathbf{u}, \mathbf{v}_j \in \mathbb{R}^n} \quad \|\mathbf{u} - \hat{\mathbf{x}}\|^2 + \sum_k \|\mathbf{v}_k\|^2 \qquad\qquad\qquad\qquad \text{(D.2)}$$

$$\text{s.t.} \quad \langle \mathbf{w}_i, \mathbf{u} \rangle - \hat{z}_i \leq \rho_i,$$

$$\sqrt{(\langle \mathbf{w}_i, \mathbf{u} \rangle - 2\hat{z}_i)^2 + \sum_k \langle \mathbf{w}_i, \mathbf{v}_k \rangle^2} - \sqrt{\langle \mathbf{w}_i, \mathbf{u} \rangle^2 + \sum_k \langle \mathbf{w}_i, \mathbf{v}_k \rangle^2} \leq 2\rho_i,$$

for all $i$, where $\mathbf{w}_i[j] = \mathbf{W}[i, j]$ is the $i$-th row of $\mathbf{W}$. Applying Lemma C.1 then rewrites (D.2) as the following.

$$\min_{\mathbf{u}, \mathbf{v}_j \in \mathbb{R}^n} \quad \|\mathbf{u} - \hat{\mathbf{x}}\|^2 + \sum_k \|\mathbf{v}_k\|^2 \quad \text{s.t.} \quad \sqrt{1 + \frac{\sum_k \langle \mathbf{w}_i, \mathbf{v}_k \rangle^2}{\hat{z}_i^2 - \rho_i^2}} \leq \frac{\langle \mathbf{w}_i, \mathbf{u} \rangle - \hat{z}}{\rho_i} \leq 1, \quad \text{(D.3)}$$

We will need the following to solve (D.3). This is the main result of this section.

**Theorem D.1** (Axial projection onto several hyperbolas). *If the problem data* $\mathbf{x} \in \mathbb{R}^m$, $\mathbf{a}_i \in \mathbb{R}^m$, $b_i \in \mathbb{R}$, $\mathbf{c}_i \in \mathbb{R}^n$ *for* $i \in \{1, 2, \ldots, \ell\}$ *satisfy*

$$\|\mathbf{C}\|^2 \cdot (\|(\mathbf{A}\mathbf{A}^T)^{-1}(\mathbf{A}\mathbf{x} - \mathbf{b})\|_\infty + \|(\mathbf{A}\mathbf{A}^T)^{-1}\|_\infty) < 1$$

*where* $\mathbf{A}[i, j] = \mathbf{a}_i[j]$, $\mathbf{b}[i] = b_i$, *and* $\mathbf{C}[i, j] = \mathbf{c}_i[j]$ *for all* $i, j$, *then the following projection*

$$(\mathbf{u}^\star, \mathbf{v}^\star) = \arg\min_{\mathbf{u}, \mathbf{v}} \left\{ \|\mathbf{u} - \mathbf{x}\|^2 + \sum_j \|\mathbf{v}_j\|^2 : (\langle \mathbf{a}_i, \mathbf{u} \rangle - b_i)^2 - \sum_j \langle \mathbf{c}_i, \mathbf{v}_j \rangle^2 \leq 1 \quad \text{for all } i \right\}$$

*has a unique solution* $(\mathbf{u}^\star, \mathbf{v}^\star)$ *with* $\mathbf{v}_j^\star = 0$.

The proof of Theorem D.1 will span the remainder of this section. Lemma 6.2 is clearly a special instance as applied to (D.3).

*Proof of Lemma 6.2.* Write $\mathbf{D}_1 = \mathrm{diag}(\rho_i)$ and $\mathbf{D}_2 = \mathrm{diag}(\sqrt{\hat{z}_i^2 - \rho_i^2})$. Then, we apply Theorem D.1 with $\mathbf{x} = \hat{\mathbf{x}}$, $\mathbf{A} = \mathbf{D}_1^{-1}\mathbf{W}$, $\mathbf{b} = \mathbf{D}_1^{-1}\hat{\mathbf{z}}$, and $\mathbf{C} = \mathbf{D}_2^{-1}\mathbf{W}$. Clearly

$$\|\mathbf{C}\|^2 = \|\mathbf{D}_2^{-1}\mathbf{W}\|^2 \leq \|\mathbf{W}\|^2/(\hat{z}_{\min}^2 - \rho_{\max}^2)$$

$$\|(\mathbf{A}\mathbf{A}^T)^{-1}(\mathbf{A}\mathbf{x} - \mathbf{b})\|_\infty = \|\mathbf{D}_1(\mathbf{W}\mathbf{W}^T)^{-1}(\mathbf{W}\hat{\mathbf{x}} - \hat{\mathbf{z}})\|_\infty \leq \rho_{\max}\|(\mathbf{W}\mathbf{W}^T)^{-1}(\mathbf{W}\hat{\mathbf{x}} - \hat{\mathbf{z}})\|_\infty$$

$$\|\mathbf{D}_1(\mathbf{W}\mathbf{W}^T)^{-1}\mathbf{D}_1\|_\infty \leq \rho_{\max}^2\|(\mathbf{W}\mathbf{W}^T)^{-1}\|_\infty$$

and hence the condition in Theorem D.1 is the following

$$\rho_{\max}\|\mathbf{W}\|^2\|(\mathbf{W}\mathbf{W}^T)^{-1}(\mathbf{W}\hat{\mathbf{x}} - \hat{\mathbf{z}})\|_\infty + \rho_{\max}^2\|\mathbf{W}\|^2\|(\mathbf{W}\mathbf{W}^T)^{-1}\|_\infty < \hat{z}_{\min}^2 - \rho_{\max}^2$$

which is the same condition stated in Lemma 6.2. $\qquad\square$

Our proof of Theorem D.1 is based on a SDP relaxation.

*Proof of Theorem D.1.* The problem is nonconvex over $\mathbf{v}$, but a convex relaxation is easily constructed by representing the quadratic outer product $\sum_k \mathbf{v}_k \mathbf{v}_k^T$ by $\mathbf{V} \succeq 0$, as in

$$\underset{\mathbf{u}\in\mathbb{R}^m, \mathbf{v}\in\mathbb{R}^n}{\text{minimize}} \quad \frac{1}{2}\|\mathbf{u} - \mathbf{x}\|^2 + \frac{1}{2}\mathrm{tr}(\mathbf{V})$$

$$\text{subject to} \quad -1 \leq \langle \mathbf{a}_i, \mathbf{u} \rangle - b_i \leq \sqrt{1 + \langle \mathbf{c}_i \mathbf{c}_i^T, \mathbf{V} \rangle} \qquad \text{for all } i \in \{1, 2, \ldots, \ell\}$$

with the relaxation being exact whenever $\mathbf{V}^\star = 0$. The corresponding Lagrangian is

$$L(\mathbf{u}, \mathbf{V}, \xi, \mu) = \frac{1}{2}\|\mathbf{u} - \mathbf{x}\|^2 + \frac{1}{2}\mathrm{tr}(\mathbf{V}) + (\xi - \mu)^T(\mathbf{A}\mathbf{u} - \mathbf{b}) - \sum_{i=1}^{\ell}\left[\xi_i\sqrt{1 + \langle \mathbf{c}_i \mathbf{c}_i^T, \mathbf{V} \rangle} + \mu_i\right],$$

over Lagrange multipliers $\xi, \mu \geq 0$. Assuming that $\mathbf{A}^T\mathbf{A} \neq 0$, this problem has strictly feasible primal point $\mathbf{u} = (\mathbf{A}^T\mathbf{A})^{-1}\mathbf{b}$ and $\mathbf{V} = \mathbf{I}$, and strictly feasible dual point $\xi = \mu = \epsilon\mathbf{1}$ for $\epsilon > 0$. Hence, strong duality is attained as in

$$\min_{\mathbf{V}\succeq 0, \mathbf{u}} \max_{\lambda, \mu \geq 0} L(\mathbf{u}, \mathbf{V}, \xi, \mu) = \max_{\lambda, \mu \geq 0} \min_{\mathbf{V}\succeq 0, \mathbf{u}} L(\mathbf{u}, \mathbf{V}, \xi, \mu).$$

Examining the inner minimization over $\mathbf{V} \succeq 0$, note that the associated optimiality conditions read

$$\nabla_{\mathbf{V}} L(\mathbf{u}, \mathbf{V}^\star, \xi, \mu) = \mathbf{S} = \frac{1}{2}\left(I - \sum_{i=1}^{q}\frac{\xi_i \mathbf{c}_i \mathbf{c}_i^T}{\sqrt{1 + \langle \mathbf{c}_i \mathbf{c}_i^T, \mathbf{V}^\star \rangle}}\right) \succeq 0, \qquad \mathbf{S}\mathbf{V}^\star = 0.$$

Hence, the minimum is attained at $\mathbf{V}^\star = 0$ if and only if $\sum_i \xi_i \mathbf{c}_i \mathbf{c}_i^T \prec I$. We will proceed to solve the dual for the optimal Lagrange multiplier $\xi^\star$ and verify that $\sum_i \xi_i^\star \mathbf{c}_i \mathbf{c}_i^T \prec I$ is satisfied.

In the case that $\mathbf{V}^\star = 0$, the corresponding $\mathbf{u}^\star$ is unique

$$\mathbf{u}^\star = \arg\min_{\mathbf{u}} L(\mathbf{u}, 0, \lambda, \mu) = \arg\min_{\mathbf{u}} \frac{1}{2}\|\mathbf{u} - \mathbf{x}\|^2 + \mathbf{y}^T(\mathbf{A}\mathbf{u} - \mathbf{b}) = \mathbf{x} - \mathbf{A}^T\mathbf{y}$$

where $\mathbf{y} = \xi - \mu$, and the dual problem is written

$$\max_{\xi, \mu \geq 0} \min_{\mathbf{u}} L(\mathbf{u}, 0, \xi, \mu) = -\min_{\mathbf{y}}\left\{\frac{1}{2}\|\mathbf{A}^T\mathbf{y}\|^2 - \mathbf{y}^T(\mathbf{A}\mathbf{x} - \mathbf{b}) + \|\mathbf{y}\|_1\right\}.$$

whose optimal conditions read

$$\mathbf{A}\mathbf{A}^T\mathbf{y}^\star - (\mathbf{A}\mathbf{x} - \mathbf{b}) \in \mathrm{sign}(\mathbf{y}^\star) \qquad \text{where } \mathrm{sign}(\alpha) = \begin{cases} +1 & \alpha > 0, \\ [-1, +1] & \alpha = 0, \\ -1 & \alpha < 0. \end{cases}$$

We wish to impose conditions on the data $\mathbf{A}, \mathbf{b}, \mathbf{x}$ to ensure that $\lambda_{\max}(\max\{0, y_i^\star\}\mathbf{c}_i\mathbf{c}_i^T) < 1$ holds at dual optimality. A conservative condition is to use the enclosure $\text{sign}(\alpha) \subset [-1, +1]$ to solve a relaxation

$$\max_{\mathbf{y}} \left\{ \lambda_{\max}\left( \sum_i \max\{0, \mathbf{e}_i^T\mathbf{y}\}\mathbf{c}_i\mathbf{c}_i^T \right) : \mathbf{y} = (\mathbf{A}\mathbf{A}^T)^{-1}(\mathbf{A}\mathbf{x} - \mathbf{b} - \mathbf{s}), \quad \mathbf{s} \in \text{sign}(\mathbf{y}) \right\}$$

$$\leq \lambda_{\max}\left( \sum_i \mathbf{c}_i\mathbf{c}_i^T \right) \cdot \max_{\mathbf{y}} \left\{ \max_i\{0, \mathbf{e}_i^T\mathbf{y}\} : \mathbf{y} = (\mathbf{A}\mathbf{A}^T)^{-1}(\mathbf{A}\mathbf{x} - \mathbf{b} - \mathbf{s}), \quad \mathbf{s} \in \text{sign}(\mathbf{y}) \right\}$$

$$\leq \lambda_{\max}\left( \sum_i \mathbf{c}_i\mathbf{c}_i^T \right) \cdot \max_{\mathbf{y}} \left\{ \max_i\{0, \mathbf{e}_i^T\mathbf{y}\} : \mathbf{y} = (\mathbf{A}\mathbf{A}^T)^{-1}(\mathbf{A}\mathbf{x} - \mathbf{b} - \mathbf{s}), \quad \|\mathbf{s}\|_\infty \leq 1 \right\}$$

$$= \lambda_{\max}\left( \sum_i \mathbf{c}_i\mathbf{c}_i^T \right) \cdot \max_i \left\{ 0, \mathbf{e}_i^T(\mathbf{A}\mathbf{A}^T)^{-1}(\mathbf{A}\mathbf{x} - \mathbf{b} - \mathbf{s}) \right\}.$$

Hence, if the following holds

$$\|\mathbf{C}\|^2 \cdot \max_i \left\{ \mathbf{e}_i^T(\mathbf{A}\mathbf{A}^T)^{-1}(\mathbf{A}\mathbf{x} - \mathbf{b}) + \sum_{j=1}^n |\mathbf{e}_i^T(\mathbf{A}\mathbf{A}^T)^{-1}\mathbf{e}_j| \right\} < 1$$

or more conservatively, if the following holds

$$\|\mathbf{C}\|^2 \cdot \left( \|(\mathbf{A}\mathbf{A}^T)^{-1}(\mathbf{A}\mathbf{x} - \mathbf{b})\|_\infty + \|(\mathbf{A}\mathbf{A}^T)^{-1}\|_\infty \right) < 1$$

then $\mathbf{V}^\star = 0$ and $\mathbf{u}^\star$ is unique as desired. $\qquad\square$

# E   Further details for multiple layers

In the general $\ell$-layer case, the SDP relaxation for problem (A) reads

$$d_{\text{lb}}^2 = \min_{\mathbf{G}_k \succeq 0} \text{tr}(\mathbf{X}_0) - 2\langle\hat{\mathbf{x}}, \mathbf{x}_0\rangle + \|\hat{\mathbf{x}}\|^2 \tag{A-lb}$$

$$\text{s.t.} \quad \begin{array}{c} \mathbf{x}_{k+1} \geq 0, \quad \mathbf{x}_{k+1} \geq \mathbf{W}_k\mathbf{x}_k + \mathbf{b}_k, \\ \text{diag}(\mathbf{X}_{k+1}) \leq \text{diag}(\mathbf{W}_k\mathbf{Y}_k^T), \\ \langle\mathbf{w}, \mathbf{x}_\ell\rangle + b \leq 0, \end{array} \quad \mathbf{G}_k = \begin{bmatrix} 1 & \mathbf{x}_k^T & \mathbf{x}_{k+1}^T \\ \mathbf{x}_k & \mathbf{X}_k & \mathbf{Y}_k \\ \mathbf{x}_{k+1} & \mathbf{Y}_k^T & \mathbf{X}_{k+1} \end{bmatrix} \succeq 0 \text{ for all } k,$$

over layer indices $k \in \{0, 1, \ldots, \ell - 1\}$, while the SDP relaxation for problem (B) is almost identical, except the constraint on $\mathbf{x}_\ell$:

$$L_{\text{lb}}^2 = \min_{\mathbf{G}_k \succeq 0} \text{tr}(\mathbf{X}_0) - 2\langle\hat{\mathbf{x}}, \mathbf{x}_0\rangle + \|\hat{\mathbf{x}}\|^2 \tag{B-lb}$$

$$\text{s.t.} \quad \begin{array}{c} \mathbf{x}_{k+1} \geq 0, \quad \mathbf{x}_{k+1} \geq \mathbf{W}_k\mathbf{x}_k + \mathbf{b}_k, \\ \text{diag}(\mathbf{X}_{k+1}) \leq \text{diag}(\mathbf{W}_k\mathbf{Y}_k^T), \\ \text{tr}(\mathbf{X}_\ell) - 2\langle\hat{\mathbf{z}}, \mathbf{x}_\ell\rangle + \|\hat{\mathbf{z}}\|^2 \leq \rho^2, \end{array} \quad \mathbf{G}_k = \begin{bmatrix} 1 & \mathbf{x}_k^T & \mathbf{x}_{k+1}^T \\ \mathbf{x}_k & \mathbf{X}_k & \mathbf{Y}_k \\ \mathbf{x}_{k+1} & \mathbf{Y}_k^T & \mathbf{X}_{k+1} \end{bmatrix} \succeq 0 \text{ for all } k,$$

over layer indices $k \in \{0, 1, \ldots, \ell - 1\}$. Note that both (A-lb) and (B-lb) are SDPs over $\ell$ smaller semidefinite variables, each of order $1 + n_k + n_{k+1}$, rather than over a single large semidefinite variable of order $1 + \sum_{k=1}^\ell n_k$. This reduction is from an application of the chordal graph matrix completion of Fukuda et al. [61]; see also [62, Chapter 10].

Now, for the choice $\hat{\mathbf{z}} = \mathbf{u} - \rho\mathbf{w}/\|\mathbf{w}\|$ where $\mathbf{u} = -b\mathbf{w}/\|\mathbf{w}\|^2$, the optimal value $L^\star$ to problem (B) gives an upper-bound to the optimal value $L^\star \geq d^\star$ of problem (A) that converges to an equality at $\rho \to \infty$. At the same time, $L_{\text{lb}} \geq d_{\text{lb}}$ holds for all $\rho > 0$ because problem (A-lb) is always a relaxation of problem (B-lb). To show this, we observe that for this choice of $\hat{\mathbf{z}}$, we have

$$\text{tr}(\mathbf{X}_\ell) - 2\langle\hat{\mathbf{z}}, \mathbf{x}_\ell\rangle + \|\hat{\mathbf{z}}\|^2 - \rho^2$$

$$= \text{tr}(\mathbf{X}_\ell) - 2\langle\mathbf{u}, \mathbf{x}_\ell\rangle + (2\rho/\|\mathbf{w}\|)\langle\mathbf{w}, \mathbf{x}_\ell\rangle + \|\mathbf{u}\|^2 - (2\rho/\|\mathbf{w}\|)\langle\mathbf{w}, \mathbf{u}\rangle + \rho^2 - \rho^2$$

$$= \underbrace{\text{tr}(\mathbf{X}_\ell) - 2\langle\mathbf{u}, \mathbf{x}_\ell\rangle + \|\mathbf{u}\|^2}_{\geq 0} + (2\rho/\|\mathbf{w}\|)[\langle\mathbf{w}, \mathbf{z}\rangle + b].$$

The nonnegativity of this first term follows because

$$\mathrm{tr}(\mathbf{X}_\ell) - 2\langle \mathbf{u}, \mathbf{x}_\ell \rangle + \|\mathbf{u}\|^2 = \mathrm{tr}(\mathbf{X}_\ell - \mathbf{u}\mathbf{x}_\ell^T - \mathbf{x}_\ell \mathbf{u}^T + \mathbf{u}\mathbf{u}^T)$$
$$= \mathrm{tr}(\mathbf{X}_\ell - \mathbf{x}_\ell \mathbf{x}_\ell^T + (\mathbf{x}_\ell - \mathbf{u})(\mathbf{x}_\ell - \mathbf{u})^T)$$
$$= \mathrm{tr}(\mathbf{X}_\ell - \mathbf{x}_\ell \mathbf{x}_\ell^T) + \|\mathbf{x}_\ell - \mathbf{u}\|^2$$

and that $\begin{bmatrix} 1 & \mathbf{x}_\ell^T \\ \mathbf{x}_\ell & \mathbf{X}_\ell \end{bmatrix} \succeq 0$ implies $\mathbf{X}_\ell - \mathbf{x}_\ell \mathbf{x}_\ell^T \succeq 0$ by the Schur complement lemma and therefore $\mathrm{tr}(\mathbf{X}_\ell - \mathbf{x}_\ell \mathbf{x}_\ell^T) \geq 0$. Hence, a feasible point $\mathbf{X}_\ell, \mathbf{x}_\ell$ for the relaxation (B-lb) satisfying $\mathrm{tr}(\mathbf{X}_\ell) - 2\langle \hat{\mathbf{z}}, \mathbf{x}_\ell \rangle + \|\hat{\mathbf{z}}\|^2 \leq \rho^2$ must immediately satisfy $\langle \mathbf{w}, \mathbf{z} \rangle + b \leq 0$ and therefore be feasible for the relaxation (A-lb).

If the relaxation (A-lb) is tight, meaning that $d^\star = d_{\mathrm{lb}}$, then the the relaxation (B-lb) must automatically be tight at $\rho \to \infty$, because $d^\star = L^\star \geq L_{\mathrm{lb}} \geq d_{\mathrm{lb}} = d^\star$. But the converse need not hold: the relaxation (A-lb) can still be loose even though (B-lb) is tight, because even with $L^\star = L_{\mathrm{lb}}$ at $\rho \to \infty$, it is still possible to have $d_{\mathrm{lb}} < d^\star$.

The nonlinear interpretation of (B-lb) reads

$$\min_{\mathbf{x}_i^{(k)} \in \mathbb{R}^p} \quad \sum_j \|\mathbf{x}_{0,j} - \hat{x}_j \mathbf{e}\|^2 \quad \text{s.t.} \quad \begin{aligned} &\langle \mathbf{e}, \mathbf{x}_i^{(k+1)} \rangle \geq \max\left\{ 0, \langle \mathbf{e}, \sum_j W_{i,j}^{(k)} \mathbf{x}_j^{(k)} + b_i^{(k)} \mathbf{e} \rangle \right\}, \\ &\|\mathbf{x}_i^{(k+1)}\|^2 \leq \langle \mathbf{x}_i^{(k+1)}, \sum_j W_{i,j}^{(k)} \mathbf{x}_j^{(k)} + b_i^{(k)} \mathbf{e} \rangle, \quad \text{for all } i, k \\ &\sum_j \|\mathbf{x}_{\ell,j} - \hat{z}_j \mathbf{e}\|^2 \leq \rho^2, \end{aligned}$$

(E.1)

over layer indices $k \in \{0, 1, \ldots, \ell - 1\}$ and neuron indices $i \in \{1, 2, \ldots, n\}$ at each $k$-th layer. Suppose that problem (B) has a trivial solution $\mathbf{x}^\star = \hat{\mathbf{x}}$ with objective zero. Then, it follows that every solution to (E.1) must be collinear and satisfy $\mathbf{x}_{0,j}^\star = \hat{x}_j \mathbf{e}$, so the relaxation (B-lb) has a unique rank-1 solution via Theorem A.2.

*Proof of Corollary 2.3.* If $\|\mathbf{f}(\hat{\mathbf{x}}) - \hat{\mathbf{z}}\| \leq \rho$, then problem (E.1) has a minimum of zero, obtained by setting $\mathbf{x}_{0,j}^\star = \hat{x}_j \mathbf{e}$ for all $j$ at the input layer. This choice of $\mathbf{x}_{0,j}^\star$ is unique, because $\sum_j \|\mathbf{x}_{0,j}^\star - \hat{x}_j \mathbf{e}\|^2 = 0$ holds if and only if $\mathbf{x}_{0,j}^\star = \hat{x}_j \mathbf{e}$, so the input layer must be collinear at optimality, meaning that $\|\mathbf{x}_{0,j}^\star\| = |\langle \mathbf{e}, \mathbf{x}_{0,j}^\star \rangle|$ for all $j$ is guaranteed to hold. Then, applying Lemma A.3 shows that $\|\mathbf{x}_{1,i}^\star - b_i^{(1)} \mathbf{e}\| = |\langle \mathbf{e}, \mathbf{x}_{1,i}^\star - b_i^{(1)} \mathbf{e} \rangle|$ and therefore $\|\mathbf{x}_{1,i}^\star\| = |\langle \mathbf{e}, \mathbf{x}_{1,i}^\star \rangle|$ for all $i$, so the first hidden layer is also collinear. Inductively repeating this argument, if the $k$-th layer is collinear, as in $\|\mathbf{x}_{k,j}^\star\| = |\langle \mathbf{e}, \mathbf{x}_{k,j}^\star \rangle|$ for all $j$, then Lemma A.3 shows that the $(k+1)$-th layer is also collinear, as in $\|\mathbf{x}_{k+1,i}^\star\| = |\langle \mathbf{e}, \mathbf{x}_{k+1,i}^\star \rangle|$ for all $i$. Hence, all solutions to (E.1) are collinear, as in $\|\mathbf{x}_{k,j}^\star\| = |\langle \mathbf{e}, \mathbf{x}_{k,j}^\star \rangle|$ for all $j, k$. Evoking Theorem A.2 then yields our desired claim. $\square$

## F  The Rank-2 Burer–Monteiro Algorithm

The Burer-Monteiro algorithm is obtained by using a local optimization algorithm to solve the nonconvex interpretation of (B-lb) stated in (E.1). In particular, fix $p = 2$, define at the $k$-th layer $\mathbf{u}_k[j] = \langle \mathbf{e}, \mathbf{x}_j^{(k)} \rangle$ and $\mathbf{v}_k[j] = \sqrt{\|\mathbf{x}_j^{(k)}\|^2 - \langle \mathbf{e}, \mathbf{x}_j^{(k)} \rangle^2}$ yields the rank-2 Burer–Monteiro problem (BM2) as desired. In turn, given a solution $\{\mathbf{u}_k^\star, \mathbf{v}_k^\star\}$ to (BM2) satisfying $\mathbf{v}_k^\star = 0$, we recover a *rank deficient* rank-2 solution to (E.1) with $\mathbf{x}_j^{(k)} = \mathbf{u}_k^\star[j]\mathbf{e}$. This rank-deficient solution is guaranteed to be globally optimal if it satisfies first- and second-order optimality; see Burer and Monteiro [56] and also [57, 58] and in particular [59, Lemma 1].

Our MATLAB implementation solves problem (BM2) using `fmincon` with `algorithm='interior-point'`, starting from an initial point selected i.i.d. from the unit Gaussian, and terminating at relative tolerances of $10^{-8}$. If the algorithm gets stuck at a spurious local minimum with $\mathbf{v}_0^\star \neq 0$, or if more than 300 iterations or function calls are made, then we restart with a new random initial point; we give up after 5 failed attempts. Empirically, we observed that whenever the SDP relaxation is tight, `fmincon` would consistently converge to a globally optimal solution satisfying $\mathbf{v}_0^\star \approx 0$ within 80 iterations of the first attempt; this suggests an underlying "no spurious local minima" result like that of Boumal et al. [57].