[Reviews · NeurIPS 2020]

Review 1

Summary and Contributions: This paper studies the tightness of SDP relaxation for one and multiple hidden layer RELU neural networks. Primarily, the "least squares" approximation (B) is studied which recovers the standard objective (A) in the limit. Most of the theoretical results are concerning conditions under which (B) is tight for a one hidden layer neural network, where the proofs are from a geometric perspective. Empirical evidence corroborates these claims.

Strengths: The proofs are quite intuitive and well explained. The approach follows primarily from the "S Lemma" strategy found in prior work, and the one hidden neuron proof was very helpful in setting the stage for the more general one hidden layer proof.

Weaknesses: I found the focus on the one hidden layer (which the majority of the body and appendices address) not extremely compelling as a result of practical import, as to the best of my understanding the semidefinite relaxations primarily focus on l >> 1. While a tight characterization is good to understand, it seems the problem is relaxed quite a bit even to understand the one hidden layer case. For instance, the linfty perturbation bound of prior work is replaced with an l2 bound (understandably, as linfty is not extremely compatible with SDP formulations), and the objective is transformed into the form (B). To this end, I was a bit confused by the setting of rho in the problem of (B). As the bounds seem to suggest that the tightness is better for rho --> infty, and simultaneously this is the setting where (B) recovers (A), why does the practitioner not take rho extremely large without loss of generality (is this due to computational considerations, or the cases where l >> 1?) To me, a question that might be worth placing more emphasis in this investigation (but was not really the focus of this paper, and not discussed greatly) was the rate of decay of tightness when the number of layers l --> very large. For instance, can we say anything nontrivial when l = 2 or 3? Understandably, this is a much more difficult question to characterize as the geometry becomes much more complicated in these cases, but it would be good to develop intuition even under more restrictions to the problem. As it stands, the paper does a good job at thoroughly exploring the l = 1 case, but the lack of discussion in any of the other cases other than it not being tight prevent me from giving a higher score. However, I should clarify that this is not really my primary area of study, so if other reviewers believe the results in l = 1 are compelling in terms of working towards a general understanding of this problem, then I can definitely be convinced to raise my score.

Correctness: I read through proofs in the l = 1 case and they seem correct to me.

Clarity: The paper was well written and easy to follow. I liked the examples and intuition given in sections 5 and 6.

Relation to Prior Work: To my understanding this is a fairly novel area of study for the setting of adversarial robustness. The explanation of the prior work [RSL18] was very helpful as a reader.

Reproducibility: Yes

Additional Feedback: I appreciate the clarifications that the authors gave, and agree that the work makes good progress towards an interesting question and am correspondingly raising my score. However, I do feel that the lack of discussion towards possible avenues of extending the current analysis to non-single layer ReLU networks makes me a bit hesitant to raise my score significantly more on the basis of how interesting the problem studied is. Understandably, extensions are substantially more difficult to analyze, but even weaker bounds or progress towards characterizing tightness in those regimes would be quite valuable in my opinion, and I encourage the authors to explore such avenues as a possible way to strengthen the paper.


Review 2

Summary and Contributions: This paper provides geometrical analysis for the tightness of the SDP relaxation of ReLUs used generally to provide adversarial robustness certificate. The authors also show that this relaxation is tight for a single hidden layer network and is usually loose for several layer networks.

Strengths: 1. The geometrical interpretation is interesting and provides good insights. 2. The hyperbolic geometry based explanation of the loose SDP relaxation for multiple layers is insightful. 3. The split of certification problems into projections on a spherical and hyperboloidal cap is insightful.

Weaknesses: Couldn’t find a major weakness. However, I would like to mention that the paper is cluttered making it a bit hard to follow.

Correctness: Yes, the method seems correct to me. Theorems make sense, however, I could not verify every line of the proof as it was a bit cluttered.

Clarity: As I mentioned, the paper is a bit cluttered. I attempted multiple times to understand the details in the paper but because of the lack of intuitions in the proofs, I could not spend enough time to verify their correctness properly.

Relation to Prior Work: Yes. This work is primarily based on Raghunathan et al. and I think the authors have credited them properly. They have also mentioned most of the prior works I am aware of.

Reproducibility: Yes

Additional Feedback: After reading the reviews, I would like to keep my old score of (6). I think an SDP relaxation based analysis of ReLU is certainly interesting and this paper does a good job in improving our understanding towards it. I would consider this paper as a step that potentially would lead to further analysis towards understanding guarantees for multiple ReLU layers (would be great if authors could provide some insight towards it).


Review 3

Summary and Contributions: The paper presents SDP relaxations to obtain bounds on the robustness of one-layer deep neural networks. The authors decompose the problem into two separate projections and apply the S-lemma or the lossy S-lemma to get tightness bounds on the relaxation. They develop clear geometrical intuitions for the relaxations. Unfortunately their results do not extend to multi-layers deep networks.

Strengths: The paper may be a first step to get certificates of robustness for deep neural networks. The overall approach, i.e. decomposing the relaxation of a non-convex problems into two parts in order to get bound may be of interest if it can be framed in a more general setting.

Weaknesses: The results do not extend to multi-layers deep-networks. Moreover the relaxation adds a radius parameter. The benefits of the theory would be much more clear if the actual problem (A) was studied. In practice, running sdp relaxations for rho varying seems not practical.

Correctness: The proofs are very well presented. Typos that require clarifications are pointed out in the comments.

Clarity: The paper is well-written. In particular numerous geometrical interpretations illustrate the approach and make the overall proofs easy to understand.

Relation to Prior Work: Yes

Reproducibility: Yes

Additional Feedback: - The authors may generalize their results easily to one conventional layer with a ReLU. This would enlarge the application of the paper. - Since the multi-layer case seems out-of-scope, could the authors explain how their approach can be beneficial as a technique to analyze sdp relaxations in two stages? Typos to clarify: - Eq 2.2.: isn't there a ||w|| missing on the right hand side? For more clarity precise that \hat z depends on rho. - Minor details: l. 138: the ReLu is a convex function. The problem is in the fact you have a composition of convex functions that is not convex. Lemma A.3: there has been a bug in this lemma (notations are incorrect). It is easy to correct but you need to do it. l. 706: there seems to be a w missing in the definition of hat z. I don't get how the next equation is obtained. Moreover I don't get the non-negativity of Tr(X) - 2<u, x> + ||u||^2. Finally the conclusion that A-lb is a relaxation of B-lb is unclear, even if z happens to ve x (note that z was not defined). ==== After rebuttal ==== The authors have not fully answered the question about the typo (u in line 706 is badly defined) but that can be considered as a detail. More importantly, I still do not understand what are the benefits of the paper for the community. It is restricted to one specific activation function on one hidden layer multi-layer perceptron network. Clearly, on the deep learning side, there is still a lot of work before making it appealing to practitioners. On the other side, robust optimization for non-linear least-squares may be quite interesting in optimization. In that case, the problem is that the paper focuses on the ReLU. THe authors should consider making a more generic proof (like a sdp for the composition of two functions, with the inner function described by LMI for example).


Review 4

Summary and Contributions: The paper “On the Tightness of Semidefinite Relaxations for Certifying Robustness to Adversarial Examples“ is devoted to a geometric study of the semidefinite relaxation of the ReLU, which leads to an algorithm for certifying robustness of a neural network to adversarial examples. In particular, the authors show that the robustness of a NN with a single hidden layer can be address in full by the semidefinite programming (e.g., SDP is tight for a least square restriction of an adversarial attack). At the same time, the situation is opposite for multiple layers. The paper is well supported theoretically. The authors correctly proved all the statements about the exactness and inexactness of the SDP relaxation. Empirically, they support the algorithm by experiments over MNIST dataset, where they use Burer-Monteiro hierarchy to solve the corresponding SDP problem. Paper is very well written, easy to read, and mathematically correct. The only concern I have is the influence on the field, but I am not an expert in neural networks and working mostly on optimization. Also, I believe that contribution over the paper “Semidefinite relaxations for certifying robustness to adversarial examples” is incremental (or, at least it should be stated more clearly), and the result is practically limited (as it applies to 1 layer NN). Overall, I like this paper but believe that it has limited influence, thus recommend "weak acceptance".

Strengths: The paper is well supported theoretically. The authors correctly proved all the statements about the exactness and inexactness of the SDP relaxation.

Weaknesses: The technique proposed by the authors is far from being practical.

Correctness: Theoretical part is correct. I have not checked the code.

Clarity: Yes, the paper is very clear and very well-written.

Relation to Prior Work: No, I believe the difference with previous works on this topic is not very well explained

Reproducibility: Yes

Additional Feedback: Dear Authors, I would appreciate if you explain more clearly the difference of their results w.r.t. to the state-of-the-art as well as the influence of it on further research regarding the robustness of neural networks. Thank you! Regards, Reviewer


Review 5

Summary and Contributions: Given a trained neural network, robustness analysis amounts to finding (or bounding) the largest perturbation to a specific input that will not cause misclassification. This problem can be written as a non-convex optimization problem, which is computationally difficult to solve exactly. To reduce computations at the expense of conservatism, convex relaxations of this problem are often considered. The authors analyze an SDP relaxation of neural network robustness certification problem, proposed by Raghunathan et al., which is based on the well-known Shor relaxation in optimization. The authors prove that this relaxation is tight for one-layer neural nets and not usually tight for multi-layer neural nets.

Strengths: The robustness certification problem is a very important problem. Most of work in this area are empirical and it is not clear whether the obtained bounds are useful or vacuous. This paper provides a theoretical result on the tightness of an SDP relaxation of this problem. From this perspective, the results of the paper are interesting and can be insightful.

Weaknesses: - The paper is not very well written and the proofs are not easy to follow, hence it does not provide much insight. There are some sentences that are vague. For example: "solving a nonconvex optimization problem from above and from below", what does from above and from below mean here?! Or "In practice, the SDP relaxation for problem (B) is often tight for finite values of the radius rho". Where is this claim coming from? Restating the main results of the paper? Furthermore, I have some clarification questions about the correctness of the results. - The authors do not motivate why instead of solving problem (A), they analyze its least-square restriction, i.e., problem (B). Having a discussion on this is important.

Correctness: - The ReLU function $z=max(0,x)$ is equivalent to $z>=0, z>=x, z^2=xz$. However, the authors use inequality instead of equality, i.e., the authors use $z>=0, z>=x, z^2<=xz$. While geometrically this will not affect the equivalence for a single neuron, it is not clear what would be the implications of using an inequality over equality on the Lagrangian relaxation, the S-procedure (or other techniques used) for the one-layer and multi-layer problem. This is a subtle issue that needs to be addressed. - Given the previous comment, it is not clear to whether the main result for single-layer neural networks is correct. If the SDP relaxation is tight, then, geometrically, one should be able to bound f(X) (X being an input set) by its convex hull through the formulation of Raghunathan et al. It would be desirable to illustrate this via 2D numerical examples. See for example, the 2D examples in "Provable defenses against adversarial examples via the convex outer adversarial polytope" by Wong and Kolter & "Safety verification and robustness analysis of neural networks via quadratic constraints and semidefinite programming" by Fazlyab, Morari, and Pappas. - In any convex relaxation of neural network robustness problem, using the pre-activation bounds has an enormous effect on the looseness of the relaxation. In the current analysis it seems that these bounds play no role in the tightness of the bounds.

Clarity: - This paper is meant to prove a result for an existing relaxation and to provide insights. However, the paper is not very well-written. The authors first provide the main results and then they discuss related work, followed by preliminaries and then the main analysis. This format was particularly hard for me to follow as I had to go back multiple times to see the main results. I would suggest the authors start with Preliminaries, then the tightness results (including the main theorems and then the analysis) for one layer and multi layers. - Why are the authors separate the last linear layer from (2.1)? Why not defining $f(x)=W_l x_l + b_l $ and then use $f_j(x) \geq f_i(x)$ as the constraint of problem (A)? This would enhance the clarity.

Relation to Prior Work: - The paper discusses an SDP relaxation proposed by Raghunathan et al. There are other SDP relaxations proposed but there is no mention to these works. For example the paper "Safety verification and robustness analysis of neural networks via quadratic constraints and semidefinite programming" by Fazlyab, Morari, and Pappas propose an SDP relaxation of the same problem and also empirically compare the tightness of the relaxation with that of Raghunathan et al.

Reproducibility: No

Additional Feedback:

[Author Response · NeurIPS 2020]

We thank the four reviewers for their careful reading, detailed feedback, and helpful comments. Below, we begin with some clarifications on our contributions and minor extensions, and end with specific responses to each reviewer.

**Why SDP?** The adversarial examples problem is relatively unique in ML in that *global optimality* matters a lot. (In training a model, this would just overfit the model.) A central weakness of attack algorithms based on local optimization (e.g. PGD of Madry et al.) is that they cannot *prove* that an adversarial example they've found is globally optimal—even if it really is globally optimal. On the other hand, provable guarantees of global optimality are certainly obtainable if we accept exp-time (e.g. Katz et al.) or exp-time in the worst-case (e.g. Tjeng et al.). Within this context, *SDP is interesting because it is the best tool for proving global optimality in poly-time.* Goemans and Williamson revolutionized combinatorial optimization when they used SDP to prove near-global optimality bounds right at the cliff edge of poly-time (assuming $P \neq NP$). Candes, Tao and their coauthors revolutionized compressed sensing when they used SDP to solve problems previous thought NP-hard to guaranteed global optimality in poly-time.

**Our contributions.** Amongst the adversarial examples literature, we are the first to guarantee a globally optimal certificate in polynomial time. Within this context, others have used SDP, and have obtained good empirical results, but we are the first to give an end-to-end theoretical proof (of any kind) for SDP. *Our paper is a first step towards global optimality in poly-time; our contribution is the proof technique to get there.* The standard technique of analyzing the SDP dual (e.g. Candes and Tao) immediately runs into painful, possibly insurmountable issues. Instead, we developed a nonconvex technique (Appendix A) that reduces the SDP into a sequence of possibly nonconvex projections. *Viewing our primary contribution as the proof technique, we have taken meticulous care in communicating the technique in a clear, clean, pedagogical way, by proving clear-cut results on simple examples.* We did this because we wanted to make it as easy as possible for future researchers to build off of our work. This is especially important because the naive SDP relaxation doesn't work well on its own, but it has the potential to be made to work well by future researchers. (Much like how Lasserre built on top of Sherali-Adams and Lovasz-Schrijver)

**Problem (A) vs Problem (B).** Our proof technique works equally well for both (A) and its *convex restriction* (B). But (A) is almost always loose, so we cannot prove anything on (A) other than to state reasons for why the SDP is loose. In comparison, problem (B) is tight for a sufficiently small $\rho$ (it is essentially a trace regularization to induce a low-rank solution) so we're able to prove predictive bounds and verify them numerically. This is a fairly common route in SDP.

**Multiple layers.** Our proof technique easily extends to the $\ell > 1$ case, as we discussed in Appendix E. The resulting problem (E.1) can be "unrolled" by recursively applying the one-layer argument. But the issue here is that after the first layer, we begin projecting onto hyperbolas. Conditions for hyperbola-on-hyperbola to be collinear can be derived but are difficult to interpret and verify (they are themselves LMIs). Nevertheless, we believe this is a direction future researchers can build off, because LMIs can always be simplified by assuming structure.

**Reviewer 1.** We would like to emphasize the fact that our result is the *first proof of tightness* within this context. Global optimality is exceedingly important within the context of adversarial examples for obvious reasons, and we give the *first provable method* that attains global optimality in poly-time. Our contribution is in the proof technique used to achieve this; we had to diverge substantially from the compressed sensing SDP literature to get here. Choice of $\ell_2$ oracle over $\ell_\infty$ oracle. Both oracles are common in the literature, but generate essentially the same adversarial examples (see e.g. Carlini and Wagner). Lipschitz constants techniques are $\ell_2$ methods that become considerably more conservative on $\ell_\infty$. SDP easily accommodate $\ell_\infty$; it is the only the theoretical analysis that becomes complicated. Large $\rho$ regime. In the regime of radius $\rho \to \infty$, we view (A) as (B) with $\hat{\mathbf{z}} = -\rho\mathbf{w}/\|\mathbf{w}\|$, but this means the center of the ball $\hat{\mathbf{z}} \to \infty$ as well. Our tightness guarantees for (B) require $\hat{\mathbf{z}}$ to remain bounded. Quantative measure of tightness. On the one-neuron example, the relaxation is tight if $|\langle \mathbf{e}, \mathbf{x} \rangle| = \|\mathbf{x}\|$. But if $|\langle \mathbf{e}, \mathbf{x} \rangle| < \|\mathbf{e}\|$, then the incidence angle $\theta = \arccos(|\langle \mathbf{e}, \mathbf{x} \rangle|/\|\mathbf{e}\|)$ gives essentially the condition number of the high-rank solution. This is an insight that only becomes clear through our proof technique; we promise to add this point to the paper.

**Reviewer 2.** We regret the cluttering noted by the reviewer. We endeavor to reduce clutter in a future revision.

**Reviewer 3.** We thank the reviewer for a number of key insights, and for catching several bugs in the Appendix. We hope our common response written above adequately addresses the reviewer's concerns regarding our contributions. Proof of Lemma A.3: agreed. Line 706 should read $\hat{\mathbf{z}} = \mathbf{u} - \rho\mathbf{w}/\|\mathbf{w}\|$. The next line should read $\text{tr}(\mathbf{X}_\ell) - 2\langle \hat{\mathbf{z}}, \mathbf{x}_\ell \rangle + \|\hat{\mathbf{z}}\|^2 - \rho^2 = \text{tr}(\mathbf{X}_\ell) - 2\langle \mathbf{u}, \mathbf{x}_\ell \rangle + \|\mathbf{u}\|^2 + 2\rho[\langle \mathbf{w}, \mathbf{z} \rangle + b]$ (the term $-\rho^2$ was lost). For nonnegativity, note that $\text{tr}(\mathbf{X}) - 2\langle \mathbf{u}, \mathbf{x} \rangle + \|\mathbf{u}\|^2 = \text{tr}(\mathbf{X} - \mathbf{u}\mathbf{x}^T - \mathbf{x}\mathbf{u}^T + \mathbf{u}\mathbf{u}^T) = \text{tr}(\mathbf{X} - \mathbf{x}\mathbf{x}^T + (\mathbf{x} - \mathbf{u})(\mathbf{x} - \mathbf{u})^T) = \text{tr}(\mathbf{X} - \mathbf{x}\mathbf{x}^T) + \|\mathbf{x} - \mathbf{u}\|^2$ but we have $\mathbf{X} - \mathbf{x}\mathbf{x}^T \succeq 0$ and therefore $\text{tr}(\mathbf{X} - \mathbf{x}\mathbf{x}^T) \geq 0$. The claim that (A-lb) is a relaxation of (B-lb) follows from the corrected version of the equation above, which shows that a feasible point $\mathbf{X}_\ell, \mathbf{x}_\ell$ satisfying $\text{tr}(\mathbf{X}_\ell) - 2\langle \hat{\mathbf{z}}, \mathbf{x}_\ell \rangle + \|\hat{\mathbf{z}}\|^2 \leq \rho^2$ would then immediately satisfy $2\rho[\langle \mathbf{w}, \mathbf{z} \rangle + b] \leq 0$. These points will be clarified and fixed.

**Reviewer 4.** We agree with the reviewer and promise to completely rewrite the introduction to better reflect our contributions. Comparison to Raghunathan et al. This previous paper was almost entirely empirical. Global optimality was not at all their focus; most of their paper was spent comparing SDP vs LP.

[Meta-Review · NeurIPS 2020]

Thank you for your submission to NeurIPS. The reviewers are generally in complete agreement that the paper provides an interesting result, that the SDP relaxation for ReLU networks is tight for one hidden layer. There was still some disagreement between the reviewers on how significant a result this truly is, though the original reviewers also had fairly low confidence in the assessment. Due to this, I solicited an additional expert review after the rebuttal and the general opinion was still the same (though the reviewer pointed out some missing literature that, while it doesn't limit the applicability of the result at all, still should definitely be mentioned in the paper). Please address these comments in the camera ready version.